# Bringing SAM to new heights: Leveraging elevation data for tree crown segmentation from drone imagery

**Mélisande Teng**[1,2] *    **Arthur Ouaknine**[1,3,4]    **Etienne Laliberté**[2]
**Yoshua Bengio**[1,2]    **David Rolnick**[1,3]    **Hugo Larochelle**[1]

[1]Mila - Québec AI Institute    [2]Université de Montréal    [3]McGill University    [4]Rubisco AI
*tengmeli@mila.quebec

## Abstract

Information on trees at the individual level is crucial for monitoring forest ecosystems and planning forest management. Current monitoring methods involve ground measurements, requiring extensive cost, time and labor. Advances in drone remote sensing and computer vision offer great potential for mapping individual trees from aerial imagery at broad-scale. Large pre-trained vision models, such as the Segment Anything Model (SAM), represent a particularly compelling choice given limited labeled data. In this work, we compare methods leveraging SAM for the task of automatic tree crown instance segmentation in high resolution drone imagery in three use cases: 1) boreal plantations, 2) temperate forests and 3) tropical forests. We also study the integration of elevation data into models, in the form of Digital Surface Model (DSM) information, which can readily be obtained at no additional cost from RGB drone imagery. We present BalSAM, a model leveraging SAM and DSM information, which shows potential over other methods, particularly in the context of plantations. We find that methods using SAM out-of-the-box do not outperform a custom Mask R-CNN, even with well-designed prompts. However, efficiently tuning SAM end-to-end and integrating DSM information are both promising avenues for tree crown instance segmentation models.

## 1   Introduction

Data on individual trees are important for understanding forest ecosystems and supporting sustainable forest management. Such data are essential, for example, to answer questions about forest composition, tree growth, and tree health and mortality. They are also particularly relevant in the context of biodiversity assessments or natural climate solutions, in measuring the carbon stored in forests and evaluating the success of afforestation, reforestation and revegetation policies [1, 2]. Specifically, access to species identity and individual tree crown delineation data is crucial, as different tree species have different allometries [3, 4, 5]. Indeed, carbon stored in a tree can be recovered with allometries using information about the crown surface area, the species and the height of the tree [6].

Individual trees are still largely monitored by conducting ground surveys [7], requiring extensive cost, time and labour. However, recent advances in deep learning, alongside the decreasing cost of drones with high-resolution cameras, open up possibilities for automatically performing individual tree crown delineation. Popular deep learning methods, such as Mask R-CNN [8] and RetinaNet [9], have been extensively used in the context of vegetation monitoring using remote sensing data, but they most often do not focus on identifying tree species [10, 11]. Despite the success of deep learning methods for tree mapping at scale using remote sensing imagery [12, 13], instance segmentation of tree crowns remains understudied, in large part because of the lack of annotated data at the individual tree level.

39th Conference on Neural Information Processing Systems (NeurIPS 2025).

In contexts where task-specific data are not abundant, practitioners often resort to pre-trained models from large datasets. The Segment Anything Model (SAM) [14], for example, is designed to segment any object in an image either in a zero-shot setting or when given prompts in the form of points, boxes, masks or text. SAM has been used out-of-the-box for a wide variety of applications, such as medical imaging [15] and river water segmentation from remote sensing imagery [16]. However, despite its zero-shot capabilities, SAM has been found to perform poorly in certain segmentation tasks when used directly in its automatic mode [17] and, consequently, a number of methods have been developed to adapt SAM to specific tasks without requiring that it be fine-tuned fully [18, 19]. In particular, RSPrompter [20] proposed to learn how to generate appropriate prompts for SAM in order to segment objects of interest in remote sensing imagery. Keeping the image encoder and mask decoder frozen, a learnable prompter taking as input the image embeddings from the image encoder is trained to produce task-relevant prompts for the mask decoder.

The integration of task-specific information from the Digital Surface Model (DSM) into tree crown instance segmentation models has also been underexplored. The DSM provides a surface elevation map including above-ground objects, and is a product that is always readily available at no additional cost from the drone RGB imagery acquisition. Indeed, Structure-from-Motion (SfM) photogrammetry, which is used to create RGB orthomosaics from high-resolution drone imagery, generates 3D photogrammetry dense point clouds, from which the DSM is derived. Thus, the DSM provides complementary 3D structural information without additional data collection overhead.

In this work, we assess the potential of SAM and the value of auxiliary DSM data for the problem of tree crown instance segmentation from high-resolution drone imagery, through three realistic use cases: boreal plantations, temperate forests and tropical forests. We introduce BalSAM, a model building on RSPrompter that allows SAM to incorporate DSM information through parameter-efficient prompt learning . We evaluate the effectiveness of BalSAM, as compared to SAM's automatic mode and RSPrompter. Our study highlights the limitations of SAM in its intended use as an out-of-the-box and user-friendly tool. However, we find that methods that learn task-specific prompts in a module integrated to SAM outperform custom-trained CNN models. We also find that integrating DSM representations within SAM or CNN-based approaches generally improves model performances for tree crown instance segmentation, with the benefits being dependent on the structural complexity of the canopy.

In summary, our contributions are: 1) assessing SAM's capacities for tree crown instance segmentation from high-resolution drone imagery, 2) introducing new methods leveraging the DSM within both SAM-based and convolutional architectures, and 3) analyzing the performances of these methods across three different forest types. This work proposes the first benchmark of instance segmentation methods on the Quebec Plantations [21], Quebec Trees [22] and BCI [23] datasets. We release project code at `https://github.com/melisandeteng/BalSAM`.

## 2   Related work

**Tree segmentation**   Recent advances in remote sensing and machine learning have enabled the mapping of trees at scale, including both detection and semantic segmentation tasks [11, 24, 10]. However, many ecological use cases (*e.g.* monitoring phenology, biomass, and species distributions) require fine-grained information on tree species and crown size, calling for instance segmentation of tree crowns by species. This task has remained understudied due to the limited availability of labelled high-resolution datasets. Brandt et al. [25] and Tucker et al. [13] successfully mapped individual trees from satellite imagery, but insufficient resolution hindered classification of tree species. In works considering tree segmentation with higher resolution data [26, 27, 28], the majority either do not classify trees or consider only a limited set of classes. Such works [29, 30, 31, 32, 33] typically rely on popular architectures such as Mask R-CNN [8] and U-Net [34], though several studies propose modified versions of Mask R-CNN to segment and classify individual tree crowns [35, 36, 37] and Firoze et al. [32] explore advanced transformer-based architectures. Classical computer vision [38, 39, 40] and machine learning [41, 42] approaches have also been explored.

**Algorithms incorporating tree height data**   Canopy height maps (CHM) derived from airborne or drone LiDAR laser returns provide complementary structural information to 2D RGB imagery and have previously been estimated [43, 44, 45, 46, 47, 48] or integrated [49] in methods developed for satellite and drone [36, 30, 50] remote sensing data. Pixel-based approaches from classical computer

vision such as watershed segmentation, region-growing and edge detection [51, 52, 53] have been used on CHM data [54] for individual tree crown delineation. However, these methods often rely on rules and careful parameter tuning, making it challenging to use them in multi-species contexts. CHM information has also been explored for individual tree crown segmentation – including relying on a custom Mask R-CNN architecture [36, 55], using the CHM as an additional input channel to a Mask R-CNN [56] or directly using raw LiDAR with point cloud-based approaches [30]. While CHMs derived from LiDAR offer high structural resolution, the digital surface model (DSM) produced by photogrammetry avoids the cost of specialized sensors and is more readily aligned with drone imagery. In addition, the DSM from photogrammetry gives an equally accurate 3D surface representation of forest canopies as LiDAR [57]. Schiefer et al. [58] found using DSM information alongside RGB drone imagery to be a promising avenue for the task of semantic segmentation of trees in drone imagery, but this work did not tackle the task of instance segmentation.

**SAM in Earth observation**   Foundation models for computer vision offer promising avenues for Earth observation tasks. In particular, the Segment Anything Model (SAM) [14] has achieved effective visual segmentation in images across a range of use cases. Several methods for adapting SAM to Earth observation have been proposed, including delineating crop field boundaries [59], classifying land cover [60] and identifying urban villages [61]. Khazaie and Wang [19] proposed a toolkit to adapt SAM to custom datasets and applied it for semantic segmentation of trees in satellite imagery. Osco et al. [18] proposed a method based on SAM, using text prompts to segment instances of a given class. However, the method requires iterative updates which would be computation and time intensive when many instances are present in an image. Further, the pre-trained text prompt encoder could be limited in its ability to capture fine-grained classes, such as different tree species. Grondin et al. [62] trained a detector to better prompt SAM to segment tree trunks from ground level imagery, but did not consider classification of species. Chen et al. [20] proposed RSPrompter, a method that learns how to generate appropriate prompts to SAM, to segment objects of interest in remote sensing imagery (see Section 4.1).

In this work, we aim to use the DSM to improve RGB-based tree crown instance segmentation, as well as enabling SAM to leverage the DSM by building upon insights from RSPrompter [20]. To our knowledge, SAM has neither been used to segment and classify individual tree crowns, nor been leveraged with height information.

## 3   Datasets

We compare methods on three datasets representing different realistic application contexts: boreal plantations, temperate forests and tropical forests. As we discuss further in Section 5, each case presents different data characteristics. Plantations (created for timber production or carbon sequestration) typically consist of trees planted in orderly rows around the same time, while forests do not, as shown in Figure 1. In this section, we present each dataset and detail the data pre-processing. Further details are presented in Appendix A.

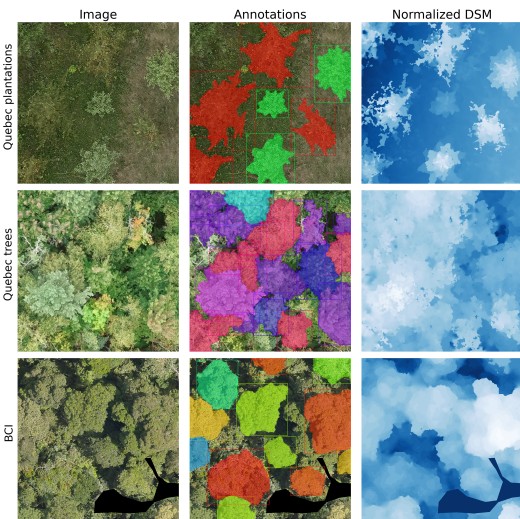

Figure 1: Examples of the raw image, annotations and DSM (normalized for the purpose of visualization) on each of the datasets under consideration.

**Quebec Plantations dataset**   We use RGB orthomosaics, photogrammetry digital surface models (DSMs), tree crown delineation and species labels in plantation sites from the UAV Canadian (Quebec) Plantations dataset [21]. The imagery has a resolution of 5 mm/pixel. We exclude the Serpentin1 and Serpentin2 sites from our study because they contain respectively only 25 and 39 annotated trees and keep 15 sites of interest.  We consider tree species that have more than 20 trees across all sites, and group the remaining species into an "Other" category, resulting in a total of 9 classes. The annotations corre-

spond to the plantations' trees, but other trees may be visible in the imagery – *e.g.* , trees outside a plantation's area on the border of the orthomosaic. We manually delineated areas of interest (AOIs) in QGIS to exclude trees that do not have a corresponding annotation in the imagery. We split the data spatially into training, validation and test sets, defining polygonal regions corresponding to geographical blocks to avoid spatial autocorrelation, and we ensure that each class is represented in all sets. Orthomosaics are either assigned entirely to a split or assigned to different splits by manually delineating areas in QGIS. We detail further the splitting strategy in Appendix A.1.

**SBL dataset**   We consider the Quebec Trees dataset [22] which covers a temperate forest site and use the RGB imagery and corresponding DSM from date 2021-09-02, for which $22,933$ tree crowns were manually labelled. In this paper, we refer to it as *SBL dataset*, for Station de Biologie de Laurentides, the site where the imagery was collected, to avoid confusion with the Quebec Plantations dataset. The resolution of the imagery is $1.9$ cm/pixel. We use the AOIs defined in Ramesh et al. [63] for training, validation and testing. Since annotations are not always available at the species level, we consider 18 classes of interest – 11 tree species, 4 genera, 2 families, a class corresponding to dead trees and an "Other" class.

**BCI dataset**   We use the 2022 imagery of the Barro Colorado Island crown maps dataset [23], covering a 50-ha rectangular plot of tropical forest at a resolution of $4$ cm/pixel with corresponding "improved version" of the crown map data. This version contains 112 species with $2,280$ tree crown delineations that were obtained by manually delineating tree crowns and further refining them with SAM with human supervision. The corresponding DSM is provided as a fourth channel to the imagery in 8-bit encoding, therefore at $1$ m-height resolution. As noted by Vasquez et al. [23], there are missing annotations from undetected tree crowns. We manually correct for missing annotations by masking out parts of the imagery that contain unannotated trees. Given the large number of species, the long-tailed distribution and challenging nature of fine-grained classification of trees in this context, we group the trees by taxonomic family. Due to the low number of instances in certain classes and the spatial split to avoid geospatial auto-correlation, we further group certain families into an "Other" class so that all families are represented in the training and test sets, leaving 31 classes of interest.

**Pre-processing**   We use the *geodataset v0.2.2*[1] Python package to divide the orthomosaics into $1024 \times 1024$ tiles with 50% overlap. We exclude tiles without labels and tiles with more than 80% black pixels at the border of the AOIs. We also exclude annotations where less than 20% of the tree appears in the tile. We detail class codes, corresponding scientific names and the number of trees per class for the different datasets in Appendix A, as well as details on the composition of the train, validation and test splits.

## 4   Methods

We extensively study the performance of SAM and the informativeness of the DSM for tree crown instance segmentation. We compare different methods, including models with the DSM used as input along with the RGB imagery and present several ablations and variations of our main methods. We detail choices of backbones and hyperparameters in Section 4.3 and Appendix B.6.

### 4.1   Methods description

**SAM out-of-the-box**   We first assess to what extent SAM can segment tree crowns in our dataset without additional training or tuning. We benchmark SAM in the automatic mask generation mode (denoted *SAM*). Following classical approaches such as watershed segmentation, we also test the use of local maxima of the DSM, potentially corresponding to treetops in the RGB image, to prompt SAM (denoted *SAM+DSM prompts*). Further details on this method in Appendix B.2. An overview of SAM+DSM prompts is shown in Figure 6 along with sample images and prompts from each dataset. For both models we apply Non-Maximum Suppression (NMS) on the segmented instances. We also considered using the DSM image as a dense prompt, but obtained very poor segmentation masks, as dense prompts are intended to be binary masks (see Appendix B.3).

---

[1]https://hugobaudchon.github.io/geodataset/index.html

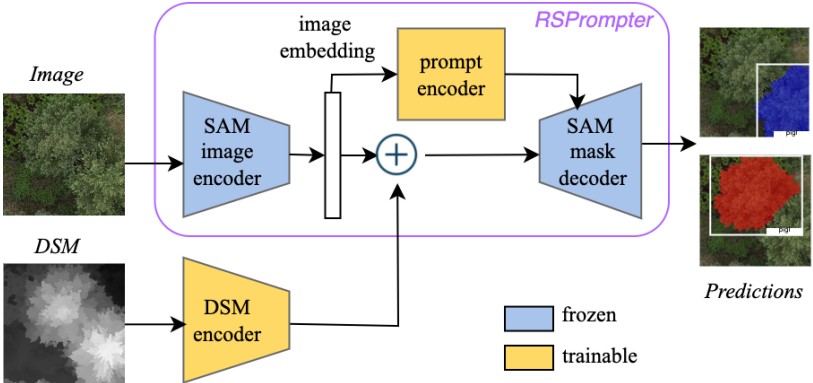

Figure 2: Overview of our BalSAM method.

**Mask R-CNN and variations** We consider Mask R-CNN as a comparison, since this architecture has previously been successfully used for tree crown instance segmentation on aerial imagery [35, 36, 11, 37]. We compare Mask R-CNN trained from scratch and initialized with weights from a model pre-trained on ImageNet.We also consider an additional variant that stacks the DSM with the RGB input as a fourth channel (*Mask R-CNN+DSM*).

**Faster/Mask R-CNN+SAM and variations** Motivated by SAM's high quality segmentation when given human input prompts, we consider training models to predict boxes and masks in an attempt to better prompt SAM and refine the predictions. We train a Faster R-CNN for tree crown detection on each dataset to provide box prompts to SAM (*Faster R-CNN+SAM*). This corresponds to the SAM-det method presented in Chen et al. [20]. We also train a Mask R-CNN on each dataset to prompt SAM with predicted boxes and/or masks (*Mask R-CNN+SAM*). Similarly to the Mask R-CNN baseline, we also consider stacking the DSM modality to its corresponding RGB image as a fourth channel for both Faster R-CNN+SAM and Mask R-CNN+SAM.

**Mask2Former** We include comparisons with the transformer-based architecture Mask2Former [64], which was developed for universal image segmentation tasks. We fine-tune Mask2Former models that were pre-trained on the COCO instance segmentation dataset [65], using the Swin-base or Swin-L backbone, and using RGB or RGB+DSM as input, stacking the DSM to its corresponding RGB image as a fourth channel.

**RSPrompter** The size of SAM makes it challenging to fully fine-tune it on small datasets. SAM's image encoder has 632M parameters and fine-tuning SAM would require considerable compute resources, rendering its training process inaccessible to most forest-monitoring practitioners. Therefore, we consider lightweight methods leveraging components of SAM without requiring full fine-tuning. We leverage RSPrompter [20] in our study, as this method was originally developed specifically for instance segmentation tasks in remote sensing imagery. We choose the RSPrompter-anchor version, as the architecture of the prompter is built on Faster R-CNN, and other methods in this benchmark are R-CNN-based. We train it following Chen et al. [20].

**BalSAM** We propose a method leveraging RSPrompter by integrating DSM embeddings to enhance SAM image representations. Our method, named **BalSAM** (in reference to the tree species balsam fir) aims at learning to better prompt SAM thanks to height information and canopy structures captured by the DSM modality. Since SAM was designed to use dense prompts with binary segmentation masks on top of point prompts, we investigate whether integrating the DSM to RSPrompter in a similar way helps guiding the segmentation and the classification. We introduce a trainable DSM encoder module to fuse DSM and image embeddings with an element-wise sum. This global embedding is then fed as input to the SAM decoder, similarly to dense prompts. An overview of BalSAM is provided in Fig. 2.

## 4.2 Evaluation

We evaluate our instance segmentation models with mean Average Precision (mAP). Given the class imbalance in our dataset, we also consider a weighted mAP (wmAP), where the weights are defined

| Model | DSM | Pre-trained | Single-class | | Multi-class | |
|---|---|---|---|---|---|---|
| | | | mAP | mIoU | mAP | wmAP |
| SAM (100 pps) | ✗ | – | 8.05 | 35.06 | – | – |
| SAM (10 pps) | ✗ | – | 10.11 | 34.01 | – | – |
| SAM | ✓(prompts) | – | 11.17 | 50.91 | – | – |
| Mask R-CNN | ✗ | ✗ | 59.36 $_{\pm 0.12}$ | 79.63 $_{\pm 0.31}$ | 42.69 $_{\pm 1.63}$ | 55.75 $_{\pm 0.87}$ |
| | ✗ | ✓ | 63.65 $_{\pm 0.25}$ | 81.82 $_{\pm 0.21}$ | 46.51 $_{\pm 0.65}$ | 58.30 $_{\pm 0.71}$ |
| | ✓ | ✓ | 64.64 $_{\pm 0.40}$ | 81.89 $_{\pm 0.35}$ | 48.96 $_{\pm 0.61}$ | 60.32 $_{\pm 0.42}$ |
| Faster R-CNN+SAM | ✗ | ✗ | 53.56 $_{\pm 0.12}$ | 76.22 $_{\pm 0.12}$ | 33.52 $_{\pm 0.25}$ | 45.79 $_{\pm 0.39}$ |
| | ✗ | ✓ | 57.85 $_{\pm 0.38}$ | 78.00 $_{\pm 0.32}$ | 39.79 $_{\pm 0.68}$ | 50.30 $_{\pm 0.87}$ |
| | ✓ | ✓ | 58.00 $_{\pm 0.14}$ | 78.27 $_{\pm 0.43}$ | 40.14 $_{\pm 0.81}$ | 52.08 $_{\pm 1.00}$ |
| Mask R-CNN+SAM | ✗ | ✓ | 57.60 $_{\pm 0.11}$ | 78.18 $_{\pm 0.18}$ | 39.76 $_{\pm 0.69}$ | 50.46 $_{\pm 0.30}$ |
| | ✓ | ✓ | 57.83 $_{\pm 0.06}$ | 77.65 $_{\pm 0.29}$ | 41.13 $_{\pm 0.65}$ | 51.33 $_{\pm 0.49}$ |
| Mask2Former (Swin-base) | ✗ | ✓ | 54.01 $_{\pm 0.70}$ | 69.80 $_{\pm 0.83}$ | 33.90 $_{\pm 0.39}$ | 42.24 $_{\pm 0.58}$ |
| | ✓ | ✓ | 58.56 $_{\pm 0.15}$ | 72.95 $_{\pm 0.21}$ | 37.36 $_{\pm 1.16}$ | 47.15 $_{\pm 5.83}$ |
| Mask2Former (Swin-L) | ✗ | ✓ | 61.77 $_{\pm 0.39}$ | 73.41 $_{\pm 0.38}$ | 44.33 $_{\pm 0.38}$ | 52.92 $_{\pm 0.49}$ |
| | ✓ | ✓ | 61.43 $_{\pm 0.4}$ | 73.38 $_{\pm 0.173}$ | 41.17 $_{\pm 0.45}$ | 51.72 $_{\pm 1.05}$ |
| RSPrompter | ✗ | – | **66.37** $_{\pm 0.53}$ | 82.58 $_{\pm 0.94}$ | 52.77 $_{\pm 0.59}$ | 62.37 $_{\pm 1.41}$ |
| BalSAM | ✓ | – | 65.03 $_{\pm 1.01}$ | **83.24** $_{\pm 0.24}$ | **54.40** $_{\pm 2.31}$ | **64.84** $_{\pm 0.86}$ |

Table 1: Results on the Quebec Plantations test dataset, averaged over 3 seeds. All metrics are multiplied by $10^2$ and reported with standard errors. The column *Pre-trained* refers to ImageNet pre-training for the backbones of the Mask R-CNN, Faster R-CNN and Mask2Former models (SAM is always pre-trained); "–" denotes not applicable. We **bold** and underline the best and second best scores.

by the proportion of examples of each class in the test set. Since SAM out-of-the-box provides segmentation masks of each instance but no associated class label, we also evaluate the models with mAP considering the single class "trees". Finally, we consider the mean Intersection over Union (mIoU) with the single class "trees", by matching each ground truth instance to the predicted instance with the highest associated IoU. We then average IoU scores over all instances in the dataset. Note that mIoU does not reflect false positive instances, as it only compares each ground truth instance with a single predicted instance – namely, the best matching one in terms of IoU. This metric reflects only the quality of the segmentation if the object has been correctly detected in a setting where we only consider a single class for all trees.

### 4.3 Implementation details

In all experiments, we use the ViT-Huge version of SAM. For the Faster R-CNN+DSM and Mask R-CNN+DSM methods, we initialize the ResNet-50 backbone of Faster R-CNN+DSM/Mask R-CNN+DSM with ImageNet weights. To allow for stacking the DSM to the image input, we randomly initialize the first layer to allow for 4 input channels. Then, we copy back the ImageNet pre-trained backbone's weights of the first layer onto the RGB channels. For all trained models, we apply RandomFlip augmentations during training and normalize the DSM by its maximum value per sample. We select the best model based on the validation segmentation mAP value (over all classes). For BalSAM, the DSM encoder follows the architecture of the dense prompt encoder in SAM and is a 3-layer CNN with layer normalization and GeLU activation. We provide further details on training hyperparameters and model architectures in App. B.6. Our methods are all trained on a single GPU with 24GB CPU memory and 48GB GPU memory.

## 5 Results

Tables 1, 2 and 3 summarize the model performances in terms of single-class "tree" metrics and aggregated mAP metrics over the classes for each dataset. We report per class mAP performance in Appendix C. The BCI dataset is the most challenging setting as it consists of a large number of classes with high visual similarity. Therefore, for this dataset, we only compared the methods that were most competitive on the Quebec Plantations and SBL datasets. We also show examples of predictions from different models in Figure 3.

| Model | DSM | Pre-trained | Single-class | | Multi-class | |
|---|---|---|---|---|---|---|
| | | | mAP | mIoU | mAP | wmAP |
| SAM (100 pps) | ✗ | – | 6.56 | 35.70 | – | – |
| SAM (10 pps) | ✗ | – | 5.63 | 21.19 | – | – |
| SAM | ✓(prompts) | – | 8.24 | 41.90 | – | – |
| Mask R-CNN | ✗ | ✗ | 26.16 $_{\pm0.35}$ | 60.07 $_{\pm0.80}$ | 19.10 $_{\pm0.23}$ | 22.45 $_{\pm0.22}$ |
| | ✗ | ✓ | 32.44 $_{\pm0.12}$ | 65.08 $_{\pm0.44}$ | 21.38 $_{\pm0.17}$ | 27.27 $_{\pm0.18}$ |
| | ✓ | ✓ | 32.37 $_{\pm0.18}$ | 64.08 $_{\pm0.17}$ | 20.87 $_{\pm0.13}$ | 26.82 $_{\pm0.15}$ |
| Faster R-CNN+SAM | ✗ | ✓ | 27.38 $_{\pm0.13}$ | 61.40 $_{\pm0.11}$ | 19.72 $_{\pm0.10}$ | 23.23 $_{\pm0.06}$ |
| | ✓ | ✓ | 28.00 $_{\pm0.09}$ | 61.49 $_{\pm0.20}$ | 20.52 $_{\pm0.10}$ | 23.89 $_{\pm0.08}$ |
| Mask R-CNN+SAM | ✗ | ✓ | 26.21 $_{\pm0.17}$ | 61.67 $_{\pm0.36}$ | 18.23 $_{\pm0.17}$ | 21.83 $_{\pm0.19}$ |
| | ✓ | ✓ | 25.94 $_{\pm0.12}$ | 61.19 $_{\pm0.17}$ | 17.73 $_{\pm0.14}$ | 21.36 $_{\pm0.10}$ |
| RSPrompter | ✗ | – | **33.59** $_{\pm1.02}$ | 64.25 $_{\pm2.64}$ | **24.94** $_{\pm0.52}$ | **29.44** $_{\pm0.83}$ |
| BalSAM | ✓ | – | 33.55 $_{\pm0.93}$ | **66.02** $_{\pm1.49}$ | 24.88 $_{\pm0.63}$ | 29.12 $_{\pm0.81}$ |

Table 2: Results on the SBL test dataset, averaged over 3 seeds. All metrics are multiplied by $10^2$ and reported with standard errors. The column *Pre-trained* refers to ImageNet pre-training for the backbones of the Mask R-CNN and Faster R-CNN models (SAM is always pre-trained); "–" denotes not applicable. We **bold** and underline the best and second best scores.

| Model | DSM | Pre-trained | Single-class | | Multi-class | |
|---|---|---|---|---|---|---|
| | | | mAP | mIoU | mAP | wmAP |
| SAM (100 pps) | ✗ | – | 8.19 | 43.13 | – | – |
| SAM (10 pps) | ✗ | – | 7.01 | 28.51 | – | – |
| SAM | ✓(prompts) | – | 11.86 | 59.76 | – | – |
| Mask R-CNN | ✗ | ✓ | 30.39 $_{\pm0.82}$ | 61.74 $_{\pm0.16}$ | 5.52 $_{\pm0.01}$ | 10.33 $_{\pm0.27}$ |
| | ✓ | ✓ | 31.93 $_{\pm0.41}$ | **63.38** $_{\pm0.79}$ | 6.34 $_{\pm0.02}$ | 10.50 $_{\pm0.23}$ |
| Mask R-CNN + DSM encoder | ✓ | ✓ | 32.62 $_{\pm0.69}$ | 63.20 $_{\pm0.68}$ | 8.30 $_{\pm0.29}$ | **11.86** $_{\pm0.27}$ |
| RSPrompter | ✗ | – | **35.55** $_{\pm0.76}$ | 60.72 $_{\pm0.85}$ | 8.44 $_{\pm0.13}$ | 11.53 $_{\pm0.34}$ |
| BalSAM | ✓ | – | 34.66 $_{\pm0.39}$ | 61.60 $_{\pm2.32}$ | **8.48** $_{\pm0.29}$ | 10.42 $_{\pm0.27}$ |

Table 3: Results on the BCI test dataset, averaged over 3 seeds. All metrics are multiplied by $10^2$ and reported with standard errors. The column *Pre-trained* refers to ImageNet pre-training for the backbones of the Mask R-CNN models (SAM is always pre-trained); "–" denotes not applicable. We **bold** and underline the best and second best scores.

## 5.1 Discussion

Overall, we find that RSPrompter and BalSAM perform better than Mask R-CNN methods and that including the DSM as additional input information improves predictions. In the following, we prioritize wmAP to assess the performance of the models–for those that can be evaluated with class-wise mAP–as our datasets have significantly unbalanced classes.

**Using SAM out-of-the-box is suboptimal, even with carefully designed prompts.** Qualitatively, we observe that in many cases, SAM automatic fails to separate overlapping crowns into separate masks and confidently segments the background or tiny plants, leading to many false positives. It also misses trees in areas where tall herbaceous vegetation occurs. We show qualitative results in Fig. 3 and Fig. 5 (App. B.1). We find that SAM+DSM, in which SAM is prompted with local maxima in the DSM, is only somewhat more performant. When a prompt corresponding to an overall treetop is given, SAM is generally able to correctly segment the tree crown, explaining the modest boost in mIoU compared to SAM automatic. However, local maxima corresponding to small plants or different parts of a single tree crown can be given as prompts to the mask decoder as shown in Fig. 7 (App. B), often leading to false positives.

Interestingly, prompting SAM with boxes or masks output by a trained Mask R-CNN degrades performance compared to the predictions of that same trained Mask R-CNN. We observe that SAM sometimes focuses on very small details and artifacts in the imagery, degrading the quality of the original segmentation. Qualitative results are shown in Figure 9 (Appendix B.4). Similarly, we find that Faster R-CNN+SAM models perform significantly worse than Mask R-CNN.

**Initializing R-CNN backbones with pre-trained ImageNet weights helps.** Mask R-CNN is competitive on all datasets, and initializing the ResNet-50 backbone with ImageNet weights of Mask R-CNN improves performance, compared to training from scratch. We make the same observation with the Faster R-CNN backbone of the Faster R-CNN+SAM method.

**State-of-the-art Mask2Former does not outperform Mask R-CNN** While we find that using RGB+DSM improves performance compared to using RGB alone as input, Mask2Former baselines (using Swin-base or Swin-L backbones) do not outperform Mask R-CNN on the task of tree crown instance segmentation on the Quebec Plantations dataset (Table 1), despite their higher capacity. This is in line with prior works that have highlighted limitations of Mask2Former in the context of forest monitoring from remote sensing imagery [66, 63].

**Methods learning to prompt SAM end-to-end outperform the other methods.** RSPrompter and BalSAM models outperform Mask R-CNN-based models (integrating or not the DSM) in terms of multi-class mAP and wmAP on all three datasets. We show qualitative results of our models' predictions on the Quebec Plantations and BCI datasets in Figure 3. Looking at class-wise metrics, we also find that RSPrompter and BalSAM generally perform significantly better than Mask R-CNN-based methods on less prevalent classes on the Quebec Plantations and SBL datasets (Table 11 in Appendix C.1 and Tables 12 and 13 in Appendix C.2).

**Integrating the DSM can improve predictions, but challenges remain for classification in dense forests with many species.** Importantly, we observe that the benefit of using the DSM is highly dependent on the structure of forested area. Intuitively, the DSM is relevant for two main reasons: (1) it captures the vertical structure of individual trees which can improve classification, (2) it represents the spatial structure of trees relative to one another, which can improve segmentation. The Quebec Plantations dataset, where the DSM impact is the greatest and most consistent across methods, is composed of well-separated young trees with visible ground. The SBL and BCI datasets are more challenging, both in terms of classification and segmentation, given the larger number of classes, overlapping tree crowns and noisy annotations. In the dense, closed canopies of the SBL dataset, individual trees hardly stand out in the DSM, as can be seen in Figure 1. The DSM is thus less informative, and models integrating the DSM perform comparably to their counterpart without DSM. We demonstrate this numerically by training a Mask R-CNN with only the DSM as input (no RGB imagery) on the Plantations and SBL datasets. We see a much larger drop in mIoU on the SBL dataset when comparing to Mask R-CNN models using image inputs, while on the Plantations dataset, mIoU remains high (see Table 14 Appendix D.1).

In the tropical forest of the BCI dataset, there are large differences between tree heights and structures, even with dense and closed canopies. Adding the DSM information improves predictions of Mask R-CNN-based models on the BCI dataset, even though it is only available at a coarse 1 m-vertical resolution. We additionally report the performance of a model encoding the DSM with a CNN module before stacking it to the RGB image and passing it as input to a Mask R-CNN (Mask R-CNN+DSM encoder in Table 3), and find that adding capacity to process the DSM information can improve further on Mask R-CNN+DSM showing great potential for future work. We provide implementation details in Appendix D.3.

**Class-wise analysis on the Quebec Plantations dataset reveals patterns aligned with challenges known to ecologists.** Looking more closely into the per-class performance for instance segmentation models (*i.e.* excluding SAM out-of-the-box based methods) on the Quebec Plantations dataset, we observe performance generally increases with the number of examples for a given class, as shown in Fig. 11 (App. C.1). However, all methods perform relatively well on *Acer saccharum* (acsa), despite there being few examples of this class, which can be attributed to this class having very different visual features than the rest of the species. The performance of the different models differs most on the *Picea mariana* (pima) class. In fact, it is a very similar species to the most common class in our dataset, *Picea glauca* (pigl). In ground field surveys, these two species are most easily distinguished by looking at the shapes of the cones rather than characteristics visible in drone imagery. In our models, incorrect classifications of *Picea mariana* tend to be for *Picea glauca* (Fig. 12 in App. C.1).

## 5.2 Ablation studies

We test several ablations and variations of our main methods using the SBL and BCI datasets.

**Mask R-CNN prompts to SAM** SAM can be prompted with both dense prompts in the form of binary masks and point prompts in the form of bounding boxes, points or text. We compare using Mask R-CNN output segmentation masks, detection boxes, or both as prompts to SAM. We find that feeding masks only yields poorer results. Additionally, computation of the mAP metric requires scores which usually correspond to detection scores for predicted boxes. We compare using boxes

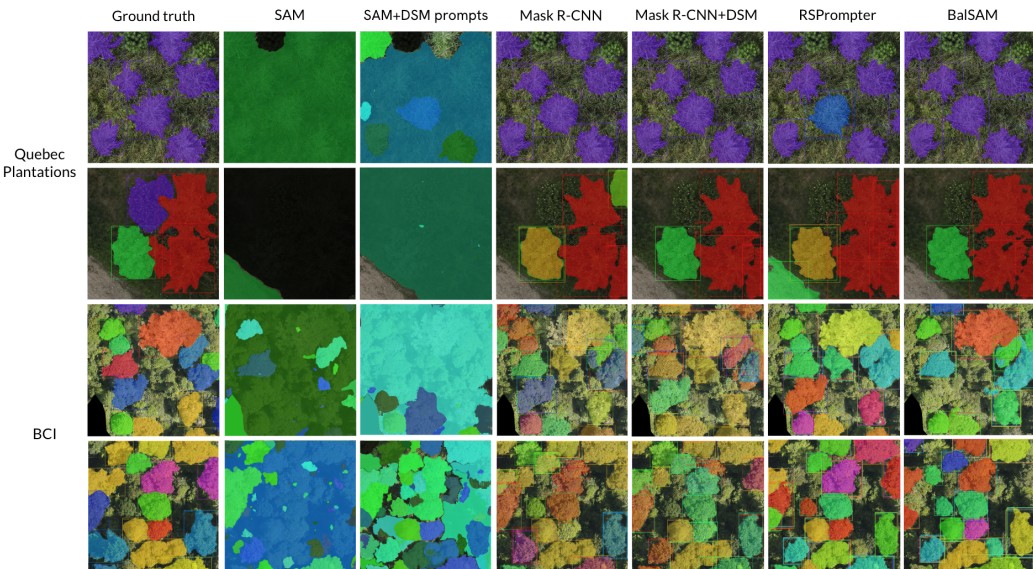

Figure 3: Qualitative results comparing methods presented in Sec. 4 on the Quebec Plantations and BCI test sets. Samples were chosen at random in the test set. For the SAM and SAM+DSM prompts columns, colours do not correspond to particular classes since SAM does not classify instances. Colours in other columns correspond to classes and are consistent across columns. BalSAM is able to produce higher quality segmentations following more closely the shape of tree crowns, and methods integrating the DSM produce fewer misclassifications.

scores from Mask R-CNN, masks scores from SAM or the average of both for the computation of the mAP. While we do not observe a significant impact on performance for different choices on the Quebec Plantations dataset, the best performance is achieved for box prompts only with boxes+masks scores on the SBL dataset (see Table 15 in Appendix D.2).

**Incorporating DSM information in Mask R-CNN**    Observing that Mask R-CNN+DSM does not perform significantly better than Mask R-CNN on the SBL dataset, we explore other ways of including the DSM. We use the DSM vertical and horizontal gradient maps as two additional channels stacked to the image input. We also consider adding capacity in the Faster R-CNN module of Mask R-CNN by adding an extra fully connected layer to the bounding box predictor and the classification head. We do not observe significance improvement in the performance as reported in Table 16 (App. D.3). Finally, we test the effect of encoding the DSM before combining it with the image – first processing the DSM through a CNN and stacking the DSM embedding to the image as a fourth channel before passing it to Mask R-CNN. We provide more details about these models in App. D.3.

**Losses**    The SBL dataset classes are highly imbalanced and we compare three losses with the standard cross-entropy used in our experiments: 1) a weighted cross-entropy loss using the inverse frequency of class occurrences in the training set as weights and 2) a hierarchical loss based on the trees taxonomy, which is a weighted sum of loss at the species, genus and family level. We define this loss in Appendix D.4.1, 3) a focal loss [9]. The gamma parameter was set to 2 for the focal loss. We find that the weighted cross entropy yields poor performance, due to the high-class imbalance, and that the model trained with focal loss does not perform as well as the cross-entropy loss. We also find that the hierarchical loss does not significantly improve performance compared to the regular cross-entropy setup (see Table 17 in Appendix D.5).

**Additional post-processing**    The default NMS in Mask R-CNN is not class agnostic, as it removes overlapping predictions only if they have the same class. Unlike in autonomous driving datasets, on which Mask R-CNN is often used and pre-trained, we do not encounter occlusions in our dataset and we expect only one object to be visible at a given location. Therefore, we consider a class-agnostic NMS, but do not observe significant improvements on the SBL or BCI datasets. This is likely because the chosen metrics favour having multiple candidate predictions for an instance – including the correct label, even if it does not have the highest score – over missing the correct class entirely.

**Variations on BalSAM**   We consider variations on how the DSM information is integrated into BalSAM. We first consider a version in which the prompt encoder receives the encoded DSM added to the image embedding as input, instead of the image embedding alone. Second, we consider a modified setup in which the mask decoder receives DSM information only through the prompt encoder. We evaluate these methods on the BCI dataset, but do not observe significant improvements from the original BalSAM model. We detail these variations in Appendix B.5.

## 5.3   Recommendations

Our study shows that using the DSM along with the RGB imagery consistently improves segmentation and classification results for the plantation use case. Therefore, we recommend that practitioners looking to quantify the carbon stored in boreal plantations include the DSM information in their models. We leave it to practitioners to decide, based on their application and available data, which models to use. For example, if only bounding box annotations are available, we showed that Faster R-CNN+SAM is a reasonable baseline, and that including the DSM helped. We also report the inference speed and the number of trainable parameters of different models in Table 10. Tropical forests remain a difficult case, with known challenges related to obtaining ground truth species labels, and to the structural and spectral similarity of forests of different taxonomic composition [67]. This poses the broader question of framing a task that meets user needs and is feasible in tropical forests, where individual tree carbon mapping might not be possible. For example, as the largest trees store the vast majority of forest carbon [68], a first step for carbon estimation in tropical forests could be to focus on large trees only, which might reduce the complexity in the number of species.

## 6   Conclusion

In this work, we investigate the potential of SAM for tree crown instance segmentation from high-resolution drone imagery, considering the settings of tree plantations, boreal forests and tropical forests. We show that methods using SAM out-of-the-box, even with well designed prompts, are suboptimal compared to the widely used architecture Mask R-CNN. However, we find that methods that learn to prompt SAM through further tuning are promising for this task. Finally, we also demonstrate that using DSM information can improve predictions. With the growing number of available drone imagery datasets for forest monitoring, the release of DSM data alongside orthomosaics may be a low-hanging fruit, as such data can be obtained directly from RGB imagery.

We highlight several limitations of the present work. On the methodological side, we find that while RSPrompter and BalSAM demonstrate superior performance to other methods, they also show higher variance. Our work does not fully address the classification challenges associated with long-tailed training data (beyond experiments with hierarchical, weighted and focal losses); further exploration through *e.g.* class rebalancing could improve performance. On the application level, we note that users building on this work should demonstrate care in regard to potential dual uses, such as risks associated with the release of models trained to identify species commonly targeted in illegal logging.

We hope our work will help advance the impactful use of machine learning in biodiversity protection and nature-based climate solutions, via improved tools for forest monitoring. Promising future directions include exploring different architectures for the DSM encoder of BalSAM, improving the methods' robustness (*e.g.* stabilising the training process with regularization and augmentations), and evaluating the effectiveness of different methods in a low-data regime or few-shot setting. Indeed, in practice, experts might be able to provide a few manual labels of species of interest. This work opens the door to other ways of using height information, for instance, predicting DSM as an auxiliary task rather than using the DSM as an input, or using the 3D point clouds obtained from SfM directly. Using a depth or canopy height map model, could also be explored. Another potential direction is adding contextual metadata into our models in the form of spatial or spectral priors, or the type of forest, to improve the classification performance. Indeed, location metadata could prove helpful, as shown in species distribution modelling contexts [69]. Spectral information, available through *e.g.* satellite open-source programs, could also be added as additional input signal into the models. Additionally, while we have so far trained models separately on each dataset, as more high-resolution drone imagery data becomes available, learning representations on combined datasets at different resolutions could provide a foundation for models that can generalize to local contexts and trees species. Developing easily adaptable methods to different forest ecosystems has considerable potential for impact.

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

## Acknowledgments

We thank Isabelle Lefebvre, Guillaume Tougas, Anne-Marie Cousineau, Chloé Fiset and members of the LEFO lab for insightful discussions and sharing field work knowledge. We thank Hugo Baudchon for iterations on the `geodataset` package and helpful discussions. We also thank Francis Pelletier and the Mila IDT team for their support with the Mila compute infrastructure.

**Funding statement** This project was undertaken partially thanks to funding from IVADO, the Canada First Research Excellence Fund, and the Canada CIFAR AI Chairs program. This research was enabled in part by compute resources, software and technical help provided by Mila (mila.quebec).

# A  Dataset

In this section, we provide more details about the composition of the datasets and the splits that we used in this study.

## A.1  Quebec Plantations dataset

We summarize the classes considered in our study in Table 4, break down the composition of each site in the dataset in Table 5 and present the distribution of species per split in Table 6.

| | |
|---|---|
| piba | *Pinus banksiana* |
| pima | *Picea mariana* |
| pist | *Pinus strobus* |
| pigl | *Picea glauca* |
| thoc | *Thuya occidentalis* |
| ulam | *Ulmus americana* |
| beal | *Betula allegnaniensis* |
| acsa | *Acer Saccharum* |
| other | Other, *Larix laricina, Pinus resinosa, Populus tremuloides, Betula papyrifera, Quercus rubra* |

Table 4: Species codes for considered classes and corresponding scientific names in the Quebec Plantations dataset.

| | piba | pima | pist | pigl | thoc | ulam | beal | acsa | other | total |
|---|---|---|---|---|---|---|---|---|---|---|
| cbpapinas | 0 | 136 | 121 | 1437 | 182 | 142 | 11 | 5 | 32 | 2076 |
| cbblackburn1 | 1440 | 215 | 102 | 100 | 0 | 0 | 1 | 0 | 7 | 1865 |
| cbblackburn2 | 573 | 18 | 50 | 993 | 0 | 1 | 0 | 0 | 67 | 1702 |
| cbblackburn3 | 0 | 0 | 0 | 140 | 3 | 0 | 0 | 0 | 2 | 145 |
| cbblackburn4 | 278 | 0 | 0 | 11 | 355 | 125 | 0 | 0 | 6 | 775 |
| cbblackburn5 | 86 | 0 | 0 | 514 | 0 | 0 | 0 | 0 | 3 | 603 |
| cbblackburn6 | 3002 | 273 | 122 | 1746 | 216 | 149 | 2 | 0 | 3 | 5513 |
| cbbernard1 | 0 | 0 | 0 | 221 | 0 | 0 | 0 | 0 | 14 | 235 |
| cbbernard2 | 0 | 0 | 14 | 61 | 0 | 0 | 0 | 0 | 0 | 75 |
| cbbernard3 | 0 | 283 | 377 | 531 | 7 | 2 | 8 | 73 | 19 | 1300 |
| cbbernard4 | 0 | 0 | 206 | 1193 | 0 | 0 | 1 | 0 | 2 | 1402 |
| afcamoisan | 0 | 0 | 0 | 628 | 0 | 0 | 0 | 0 | 2 | 630 |
| afcahoule | 0 | 0 | 0 | 1004 | 0 | 0 | 0 | 0 | 1 | 1005 |
| afcagauthmelpin | 0 | 0 | 0 | 0 | 0 | 0 | 0 | 0 | 1674 | 1674 |
| afcagauthier | 0 | 0 | 0 | 500 | 0 | 0 | 0 | 0 | 0 | 500 |
| Total | 5379 | 925 | 992 | 9079 | 763 | 419 | 23 | 78 | 1842 | 19500 |

Table 5: Number or trees per species per site of the Quebec Plantations dataset.

The training, validation and test sets were defined such that all classes were represented in the training and train and test set. There are neither image pixels, nor annotations that belong to two different splits. While some sites are represented in both the testing and training sets in the Quebec Plantations dataset, we tried as much as possible to assign sites fully to splits and limit spatial autocorrelation. The only reason why some orthomosaics were divided into both the training and test sets was ensuring that species of interest were both in the train and test sets. This is the only time when we manually drew AOIs. However, as much as possible, the train/val/test regions were kept as spatially separate

| | piba | pima | pist | pigl | thoc | ulam | acsa | beal | other | **total** |
|---|---|---|---|---|---|---|---|---|---|---|
| train | 19869 | 1377 | 2224 | 32496 | 1079 | 709 | 179 | 51 | 3343 | 61327 |
| val | 6978 | 2046 | 2447 | 6710 | 573 | 245 | 116 | 40 | 3713 | 22868 |
| test | 1471 | 1056 | 544 | 6519 | 1946 | 1050 | 56 | 19 | 1601 | 14262 |

Table 6: Tree species annotations distribution in the different Quebec Plantations splits. Note the values for each set and species are higher than the number of trees because tiles have 50% overlap.

| Site | Train | Val | Test |
|---|---|---|---|
| cbpapinas | | ✓ | ✓ |
| cbblackburn1 | | ✓ | |
| cbblackburn2 | ✓ | | |
| cbblackburn3 | | | ✓ |
| cbblackburn4 | | | ✓ |
| cbblackburn5 | ✓ | | |
| cbblackburn6 | ✓ | | |
| cbbernard1 | ✓ | | |
| cbbernard2 | | ✓ | |
| cbbernard3 | | ✓ | ✓ |
| cbbernard4 | ✓ | | |
| afcamoisan | ✓ | | |
| afcahoule | ✓ | | |
| afcagauthmelpin | ✓ | ✓ | ✓ |
| afcagauthier | ✓ | ✓ | ✓ |

Table 7: Assignment of sites of the Quebec Plantations dataset to splits.

"blocks" ensuring no overlap between the splits. Table 7 shows site assignment to splits. Only the sites afcagauthmelpin and afcagauthier were split into the train and test sets.

## A.2 SBL dataset

While Ramesh et al. [63] and Cloutier et al. [22] conducted previous studies on the SBL dataset in the context of semantic segmentation, we modify some of the classes used these studies, noting that some classes in the annotations were ignored. As much as possible, we group those classes into ones already considered in the study. Some annotations are only provided at the genus or family level. For example, classes of interest include Acer saccharum, Acer rubrum and Acer pensylvanicum. but some instances only have the label "Acer". We choose to keep genus level classes as separate classes instead of grouping them all into an "Other" category as it would end up being composed of many different species. Certain species only have very few instances and we group them into two supercategories "Pinopsida" and "Magnoliopsida" for conifers and non-conifers, which are also the level at which some annotations are provided.

We summarize the classes we consider for the task on the SBL dataset, with the corresponding names in the original annotations in Table 8. We also show the number of instances per class per split in Table 9.

## A.3 BCI dataset

The BCI dataset contains annotations for 2280 tree crowns covering 112 species. Given the long-tailed distribution of tree species and the need to split the orthomosaic spatially to avoid spatial auto-correlation, we decide to consider classes at the family level. We group further group some families into the "Other" class, such that all families are present in the train and test sets. We show the distribution of trees from the BCI dataset family classes considered in our study in 4. The "Other" class contains the following families: Clusiaceae, Polygonaceae, Malpighiaceae, Myrtaceae, Erythropalaceae, Vochysiaceae, Erythroxylaceae, Sapindaceae, Staphyleaceae, Lythraceae, Elaeocarpaceae, Rhizophoraceae, Monimiaceae, Violaceae, Solanaceae and Other.

| Class | Corresponding annotation codes |
|---|---|
| Dead | *Dead* |
| Pinopsida | *Conifere* |
| Magnoliopsida | *Feuillus, QURU (Quercus rubra L.), OSVI (Ostrya virginiana (Mill.) K.Koch), PRPE (Prunus pensylvanica L.fil.), FRNI (Fraxinus nigra Marshall)* |
| Thuja occidentalis L. | *THOC (Thuja occidentalis)* |
| Abies balsamea (L.) Mill. | *ABBA (Abies balsamea)* |
| Larix laricina (Du Roi) K.Koch | *LALA (Larix laricina)* |
| Tsuga canadensis (L.) | *TSCA (Tsuga canadensis)* |
| Betula L. | *Betula, BEPO (Betula populifolia Marshall)* |
| Fagus grandifolia Ehrh. | *FAGR (Fagus grandifolia)* |
| Populus L. | *Populus, POBA (Populus balsamifera L.), POGR (Populus grandidentata Michx), POTR (Populus tremuloides Michx.)* |
| Acer L. | *Acer* |
| Acer pensylvanicum L. | *ACPE (Acer pensylvanicum)* |
| Acer saccharum Marshall | *ACSA (Acer saccharum)* |
| Acer rubrum L. | *ACRU (Acer rubrum)* |
| Pinus strobus L. | *PIST (Pinus strobus)* |
| Betula alleghaniensis Britton | *BEAL (Betula alleghaniensis)* |
| Betula papyrifera Marshall | *BEPA (Betula papyrifera)* |
| Picea A.Dietr. | *Picea, PIGL (Picea glauca (Moench) Voss), PIMA (Picea mariana (Mill.) Britton et al.), PIRU (Picea rubens Sarg.)* |

Table 8: Classes considered in the SBL dataset, as well as corresponding codes and scientific names in the original annotations. In orange are classes at the family level, and in teal are classes at the genus level.

| | dead | Pinopsida | Magnoliopsida | THOC | ABBA | LALA | TSCA | Betula | TAGR. | Populus | Acer | ACPE | ACSA | ACRU. | PIST | BEAL | BEPA | Picea | **Total** |
|---|---|---|---|---|---|---|---|---|---|---|---|---|---|---|---|---|---|---|---|
| Train | 2434 | 21 | 389 | 3160 | 5174 | 481 | 37 | 8 | 363 | 4423 | 1138 | 2297 | 3905 | 17693 | 2102 | 289 | 19474 | 1367 | 64755 |
| Val | 642 | 68 | 169 | 1561 | 2970 | 6 | 129 | 12 | 125 | 952 | 466 | 81 | 330 | 3746 | 680 | 673 | 3632 | 1101 | 17343 |
| Test | 800 | 149 | 76 | 1964 | 4622 | 282 | 92 | 0 | 582 | 403 | 538 | 1081 | 803 | 5934 | 129 | 485 | 5125 | 1958 | 25023 |

Table 9: Number of instances per class per split in the SBL dataset. Note that the number of instances is higher than the number of trees since we have overlapping tiles.

# B  Models

## B.1  SAM automatic

We provide more examples of predictions of SAM in its automatic mode on the Quebec Plantations dataset in Figure 5.

## B.2  SAM+DSM prompts

In Figure 6, we show an overview of the SAM+DSM prompts method described in Section 4. Details on how local maxima are obtained are provided in Appendix B.6. Figure 7 shows some examples of local maxima that are fed as prompts to the mask decoder. One limitation of this method is in the case where there are a lot of small plants sticking out of the ground, giving many local maxima prompts that do not correspond to a tree (third row of the Quebec Plantations column). The case of SBL shows that manual tuning of a single neighborhood size parameter to define the height prompts has its limitations. While the chosen parameter is suited for areas with smaller trees (rows 1 and 2), it leads to many prompts on the same object for larger trees. The many prompts on the BCI dataset images are due to the fact that the DSM is only given at 1 m-height resolution, so clustered points often correspond to points with the same DSM value. Note that having too many DSM prompts on tree crowns even if they are not in the center is not a major limitation of this method, aside from computation time. Indeed, each point is fed independently to SAM, and we apply NMS to the predictions, so if segmented objects overlap, only the object with highest confidence score is kept. While there are many ways to manually refine rules for filtering local maxima, we did not do any filtering. We originally experimented with setting thresholds on the DSM or tuning further the size of

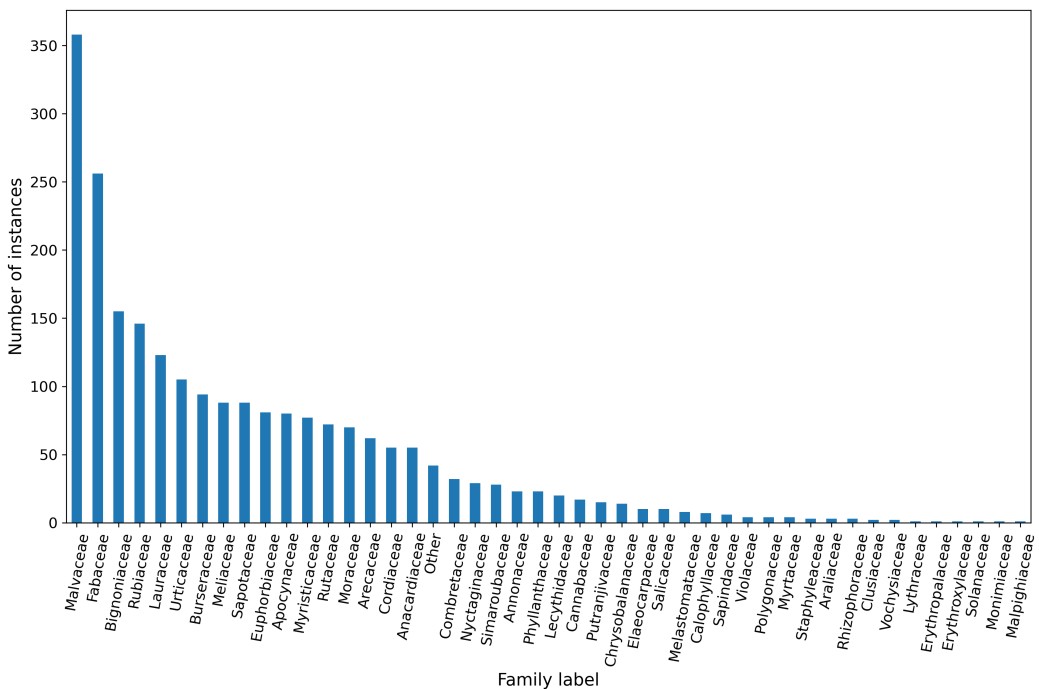

Figure 4: Distribution of trees of each of the considered families in the BCI dataset, ordered by decreasing prevalence.

the neighbourhood to define local maxima. However, there is tall herbaceous vegetation in some sites, and well-tuned parameters on a site would not necessarily transfer well to other sites within the same dataset. Also, the DSM does not provide height with respect to the ground but rather relative height between objects in the image, and does not account for differences in terrain elevation. While it could be possible to normalize the DSM with respect to the lowest point in a site, if we have imagery from a site on a very angled slope, filtering local maxima based on a threshold would not necessarily bring much improvement.

### B.3 SAM+DSM mask prompts

We also tried feeding a normalized DSM to SAM as a mask prompt. SAM normally calls for binary mask prompts, and feeding the DSM as a mask prompt would give gridded segmentations which were not satisfactory enough to be included in this study, as shown in Figure 8.

### B.4 Mask R-CNN+SAM

Figure 9 shows examples of predictions of Mask R-CNN and Mask R-CNN+SAM (in which boxes and masks of the former are fed to SAM). While SAM can refine Mask R-CNN segmentations successfully in some cases (see first row), it also leads to gridded segmentation patterns, derading the segmentations overall (second and third row).

### B.5 BalSAM variations

We also propose variations on BalSAM, using the addition of the image embedding and the output of the DSM encoder as input to the prompt encoder. We show overviews of these variations in Figure 10.

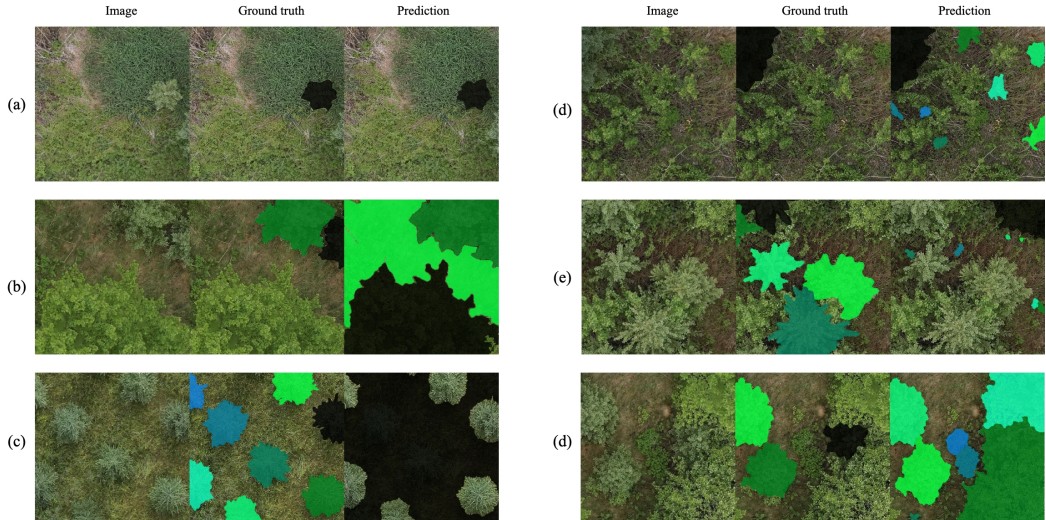

Figure 5: Examples of SAM automatic predictions: (a) A success case. (b) SAM segments everything, including the background, and merging two touching crowns into a single instance in the top right corner. (c) SAM segments only the background, i.e., everything but the objects of interest. (d) A lot of tiny isolated objects are segmented. (e) SAM completely misses the objects of interest. (f) SAM segments the trees and also the large bushes around.

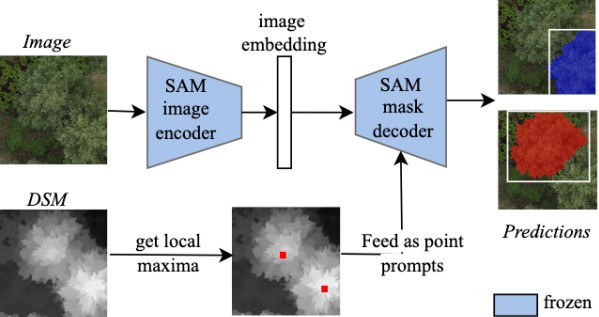

Figure 6: Overview of our SAM+DSM prompts method.

## B.6 Implementation details

We first evaluate SAM in its automatic mode on the test set tiles with a points per side (pps) value of 100 (default parameter) and 10. For SAM+DSM prompts, the local maxima in the DSM are obtained with `skimage.feature.peak_local_max` function, setting the parameter for minimal allowed distance separating peaks to 50 for the Quebec Plantations dataset and 20 for the SBL and BCI datasets. We also tried using `scipy.ndimage.maximum_filter` to find the local maxima but this led to poorer performance. For all SAM out-of-the-box methods, NMS is applied on the predictions with a score threshold of 0.5 and overlap IoU threshold of 0.5.

All Mask R-CNN-based models use the `torchvision` implementation of Mask R-CNN and are trained with SGD optimizer with learning rate 0.0001, momentum 0.9, and weight decay 0.0005, and linear warmup starting at $10^{-6}$. Models are trained for a maximum of 100, 300, and 500 epochs on the Quebec Plantations, SBL and BCI datasets respectively. Batch size is 32 for Mask R-CNN and 8 for Mask R-CNN+DSM. NMS is applied with the default parameters. We initialize the ResNet-50 backbone of Mask R-CNN+DSM with ImageNet weights, and for the first layer, copy the weights to the channels corresponding to the RGB input.

All Faster R-CNN-based models use the `torchvision` implementation of Faster R-CNN are trained for a maximum of 100 epochs on the Quebec Plantations dataset, and Adam optimizer with learning

Quebec plantations          SBL          BCI

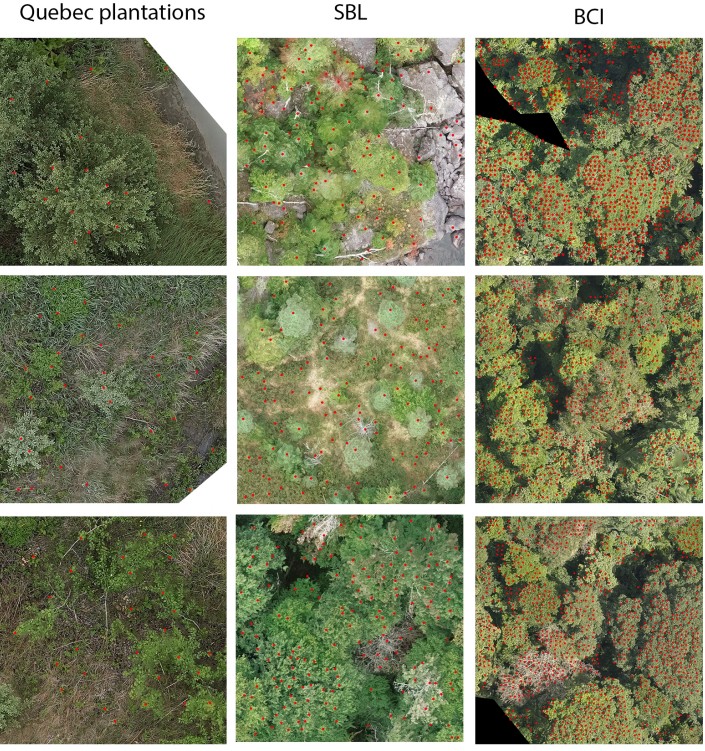

Figure 7: Examples of images with overlayed local maxima prompts for the Quebec Plantations (left column), SBL (middle column) and BCI (right column) datasets.

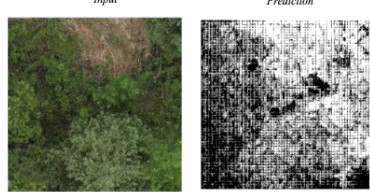

Figure 8: Examples of image and prediction when the DSM is fed as a mask prompt to SAM.

rate 0.0001 for finetuning and 0.0005 when trained from scratch, betas of 0.9 and 0.999, weight decay of 0.0005, and using an exponential decay scheduler updating the learning rate each 10 epochs. Batch size is 32 for Faster R-CNN and 16 for Faster R-CNN+DSM. NMS is applied with the default parameters. We initialize the ResNet-50 backbone of Faster R-CNN+DSM with ImageNet weights, and for the first layer, copy the weights to the channels corresponding to the RGB input.

For Faster R-CNN+SAM and Mask R-CNN+SAM methods, the scores used to compute the mAP metrics are the average of the output scores of Faster R-CNN/Mask R-CNN and SAM predicted IoU scores.

For the Mask2Former models, we used the *Swin-base* and the *Swin-large* versions of the model, and initialized the models with COCO-pretrained weights, using the implementation available through Transformers. The default preprocessing and postprocessing for evaluation were left unchanged. Batch size is 8 for the Swin-base and 4 for Swin-large Mask2Former models. Swin-base models are trained for a maximum of 100 epochs, Swin-large models are trained for a maximum of 50 epochs, as these models tend to overfit on our datasets.

Following Chen et al. [20], the RSPrompter based methods are trained with input images of size 1024×1024, normalized with ImageNet statistics, and learning rate scheduler strategy of linear

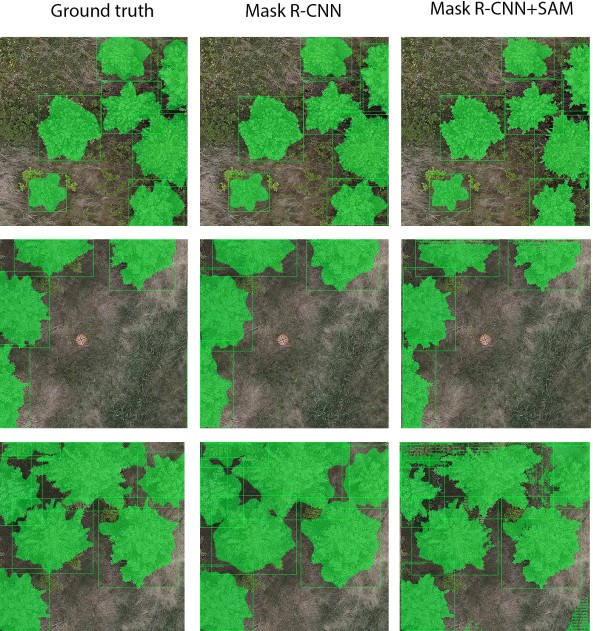

Figure 9: Examples of ground truth, Mask R-CNN and Mask R-CNN+SAM predictions on the Quebec Plantations dataset.

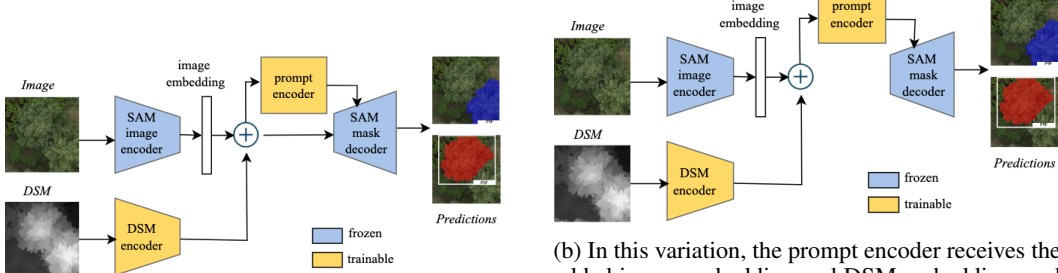

(a) In this variation, the prompt encoder receives the added image embedding and DSM embedding.

(b) In this variation, the prompt encoder receives the added image embedding and DSM embedding, and no prompt is fed in the dense prompt branch of the mask decoder.

Figure 10: Variations on BalSAM

warmup followed by cosine annealing. The models are trained with batch size 2 (as in Chen et al. [20]'s experiments), base learning rate of 0.00001 with linear warmup starting at $10^{-8}$ for one epoch followed by cosine annealing. We use AdamW optimizer with weight decay 0.1. Models are trained for a maximum of 50, 100 and 200 epochs on the Quebec Plantations, SBL and BCI datasets respectively. The DSM encoder of BalSAM is a 3-layer CNN with layer normalizations and GeLU activations. The CNN layers are defined as following:

- First layer: Kernel size $(2, 2)$, with 192 output channels, and a stride of $(2, 2)$.

- Second layer: Kernel size $(8, 8)$, with 768 output channels, and a stride of $(8, 8)$.

- Third layer: Kernel size $(1, 1)$, with 256 output channels, and a stride of $(1, 1)$.

All the models were trained on a single RTX8000 or A100 GPU, requiring up to 48GB GPU memory and 24GB CPU memory. Only the Mask R-CNN+DSM encoder model and the variations on BalSAM, presented in the Ablations required a larger GPU, and were trained on a A100 GPU with 80GB GPU memory and 48GB CPU memory. Experiments took between one and five days to complete, depending on models and batch size.

While RSPrompter and BalSAM use a batch size of 2 due to the large input image size of 1024x1024 pixels, following [20], and the models were left to train until the maximum number of epochs was reached, the best model (selected with the validation set mAP) is usually trained for fewer epochs than Mask R-CNN-based models.

We report average inference speed per sample for different models using the same V100-SXM2-32GB GPU in Table 10. While we did not control for the type of GPU used across experiments and therefore training cost figures would be misleading, we report the number of trainable parameters in each model (this includes the final layer for the Plantations dataset, which has 9 classes). For models that can also take the DSM as a 4th channel along the RGB images as input, the addition of the DSM does not lead to a significant increase in number of parameters.

| Model | Inference speed (s/image) | Number of trainable parameters |
|---|---|---|
| SAM automatic | 15.3 | 0 |
| SAM+ height prompts | 1.97 | 0 |
| Mask R-CNN | 0.34 | 43.7M |
| Mask2Former Swin-base* | 0.27 | 106.9M |
| Mask2Former Swin-large* | 0.30 | 215.5M |
| RSPrompter | 1.00 | 117M |
| BalSAM | 1.04 | 126M |

Table 10: Comparison of average inference speed on the Plantations test dataset and number of trainable parameters of different models (*note that Mask2Former models used images that are 384x384 pixels inputs when all the other models used 1024x1024)

.

## C  Results

### C.1  Class-wise performance on the Quebec Plantations dataset

Figure 11 shows the per-class mAP performance of different models on the Quebec Plantations test set.

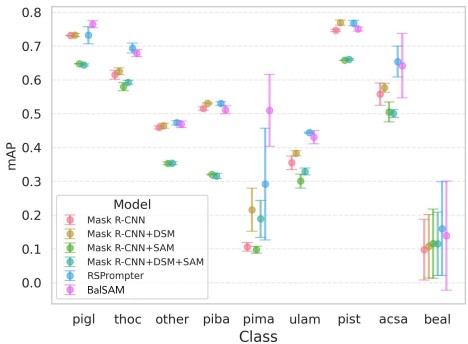

Figure 11: Per class mAP performance on the Quebec Plantations test set. For each model, the performance is averaged on 3 seeds and we show standard deviations. Tree species on the x-axis are ordered by decreasing prevalence in the dataset from left to right. Mask R-CNN is pre-trained on ImageNet. Numerical results are provided in Table 11.

In Table 11, we report the per class mAP on the test set for the different methods in our study, to the exception of the SAM out-of-the-box methods which do not classify the predicted masks. We also show a confusion matrix for predictions of a Mask R-CNN model on the Quebec Plantations dataset in Figure 12.

### C.2  Class-wise performance on the SBL dataset

We report per-class mAP on each of the classes in the SBL test set in Tables 12 and 13. We order classes in the tables by decreasing prevalence in the training set. We observe that RSPrompter and

| Model | DSM | pre-trained | piba | pima | pist | pigl | thoc | ulam | other | beal | acsa |
|---|---|---|---|---|---|---|---|---|---|---|---|
| Mask R-CNN | ✗ | ✗ | 45.88 ±0.08 | 13.31 ±5.87 | 73.60 ±0.30 | 69.75 ±0.63 | 60.08 ±0.43 | 30.90 ±1.35 | 41.82 ±0.12 | 7.64 ±6.80 | 41.23 ±3.67 |
| Mask R-CNN | ✗ | ✓ | 51.55 ±0.29 | 10.59±0.75 | 74.67 ±0.20 | 73.14 ±0.02 | 61.53 ±0.77 | 35.49 ±1.15 | 45.94 ±0.28 | 9.81 ±5.18 | 55.78 ±1.88 |
| Mask R-CNN | ✓ | ✓ | 53.08 ±0.10 | 21.56 ±3.68 | 76.97 ±0.43 | 73.29 ±0.29 | 62.57 ±0.61 | 38.30 ±0.43 | 46.43 ±0.40 | 10.78 ±5.42 | 57.69 ±0.76 |
| Faster R-CNN+SAM | ✗ | ✗ | 60.56 ±0.05 | 56.28 ±0.51 | 28.96 ±0.53 | 24.53 ±0.81 | 3.32 ±0.31 | 26.26 ±1.13 | 60.51 ±0.84 | 41.2 ±3.97 | 0.0 ±0.0 |
| Faster R-CNN+SAM | ✗ | ✓ | 64.11 ±0.65 | 61.03 ±0.80 | 34.93 ±0.59 | 31.57 ±0.37 | 3.61 ±2.28 | 33.58 ±2.37 | 66.65 ±0.10 | 49.82 ±2.30 | 12.84 ±2.90 |
| Faster R-CNN+SAM | ✓ | ✓ | 66.04 ±0.68 | 61.38 ±0.27 | 35.46 ±0.06 | 30.78 ±0.18 | 17.26 ±7.86 | 31.94 ±0.39 | 66.37 ±0.35 | 51.86 ±0.87 | 0.15 ±0.15 |
| RSPrompter | ✗ | – | 53.03 ±0.29 | 29.17 ±9.53 | 76.83 ±0.47 | 73.23 ±1.45 | 69.43 ±0.85 | 44.40 ±0.12 | 47.33 ±0.37 | 16.00 ±7.99 | 65.43 ±2.65 |
| BalSAM | ✓ | – | 51.17 ±0.69 | 50.97 ±6.15 | 75.07 ±0.35 | 76.47 ±0.61 | 67.93 ±0.57 | 43.07 ±1.12 | 46.87 ±0.54 | 13.93 ±9.32 | 64.23 ±5.53 |

Table 11: mAP per class $[10^2]$ with standard errors on the Quebec Plantations test set for the instance segmentation models in our study.

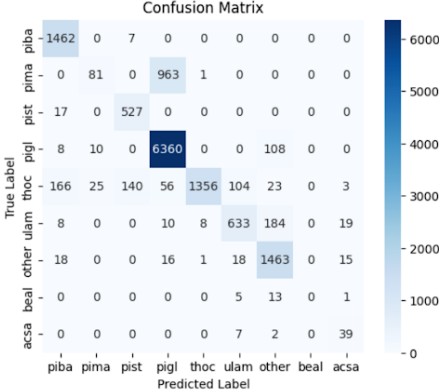

Figure 12: Confusion matrix for the Mask R-CNN model predictions on the Quebec Plantations test set. *Picea mariana* (pima) is most often confused with *Picea glauca* (pigl), which is a very similar looking species, and the most common in the dataset.

BalSAM significantly improve on Mask R-CNN methods on less common classes in the dataset, in particular for Picea, Fagus grandifolia (FAGR) and Tsuga canadensis (TSCA) classes.

| Model | BEPA | ACRU | ABBA | Populus | ACSA | THOC | Dead | ACPE | PIST |
|---|---|---|---|---|---|---|---|---|---|
| Mask R-CNN | 33.87 ± 0.19 | 21.84 ± 0.12 | 35.59 ± 0.13 | 40.47 ±0.08 | 19.64 ±0.48 | 29.65 ±0.46 | 18.30 ± 0.66 | 9.93 ± 0.72 | 44.63 ± 0.14 |
| Mask R-CNN+DSM | 33.61 ± 0.02 | 21.74 ± 0.25 | 35.31 ± 0.39 | 41.55 ± 0.08 | 17.99 ± 0.27 | 28.68 ± 0.45 | 17.85 ± 0.03 | 9.72 ± 0.33 | 44.90 ± 1.48 |
| RSPrompter | 34.98 ± 0.67 | 23.08 ± 0.64 | 36.48 ± 1.10 | 43.77 ± 0.50 | 20.92 ± 0.85 | 33.14 ± 0.47 | 22.63 ± 0.81 | 10.90 ± 0.64 | 48.12 ± 2.17 |
| BalSAM | 34.76 ± 1.02 | 22.12 ± 0.75 | 36.53 ± 0.43 | 45.73 ± 1.52 | 20.96 ± 1.05 | 32.58 ± 0.73 | 21.38 ± 1.71 | 10.46 ± 0.25 | 47.53± 1.46 |

Table 12: Per-class mAP on the SBL dataset for the most prevalent classes in the training set (ordered from left to right in decreasing order of prevalence)

| Model | Picea | Acer | LALA | Magnoliopsida | FAGR. | BEAL | TSCA | Pinopsida |
|---|---|---|---|---|---|---|---|---|
| Mask R-CNN | 31.93 ±0.75 | 0.00 ± 0.01 | 40.23 ± 1.69 | 0.00 ±0.00 | 15.09 ± 0.69 | 21.66 ±0.35 | 0.53 ± 0.43 | 0.00 ±0.00 |
| Mask R-CNN+DSM | 30.92 ±0.98 | 0.00 ±0.00 | 40.76 ±1.09 | 0.00 ±0.00 | 11.68 ±0.15 | 20.11 ±1.39 | 0.00 ±0.00 | 0.00 ±0.00 |
| RSPrompter | 40.83 ±0.50 | 1.01 ±0.30 | 43.92 ±0.79 | 0.86 ±0.43 | 18.17 ±1.65 | 21.67 ±1.29 | 23.50 ±1.88 | 0.02 ±0.01 |
| BalSAM | 40.73 ±0.96 | 1.11 ±0.05 | 45.13 ±1.15 | 0.25 ±0.07 | 17.69 ±0.09 | 22.62 ±0.93 | 23.38 ±3.68 | 0.00 ±0.00 |

Table 13: Per-class mAP on the SBL dataset for the least prevalent classes in the training set (ordered from left to right in decreasing order of prevalence)

# D    Ablations

## D.1    Informativeness of the DSM

We trained a Mask R-CNN with DSM inputs only (no RGB image) to assess its informativeness on the Plantations dataset vs the SBL dataset and report results in Table 14. When looking at the drop in performance compared to the Mask R-CNN models using RGB images as inputs, we observe a

much larger drop on the SBL dataset compared to the Plantations dataset. In fact, single-class mIoU remains high, which is another way to see that tree crowns are distinguishable from the background and from each other using the DSM only, and points to the usefulness of the DSM in plantation contexts.

| Dataset | single-class mAP | mIoU | mAP |
|---|---|---|---|
| Plantations | 0.464 | 0.734 | 0.178 |
| SBL | 0.092 | 0.289 | 0.023 |

Table 14: Test set performance of Mask-R-CNN (pre-trained on ImageNet) using DSM only as input

.

## D.2 Mask R-CNN+SAM prompts and scores

We explore using mask predictions, box predictions or both, output by a trained Mask R-CNN, as prompts to SAM. Additionally we consider different prediction scores, using either box scores only from the Mask R-CNN, or the average of the IoU scores of SAM and the box scores of the Mask R-CNN for computing the evaluation metrics. We report performance on the SBL dataset for different combinations of scores and prompts in Table 15.

| | box prompts | mask prompts | box score | mask+box score | Single-class | | Multi-class | |
|---|---|---|---|---|---|---|---|---|
| | | | | | mAP | mIoU | mAP | wmAP |
| Mask R-CNN+SAM | ✓ | | ✓ | | **24.59** ±0.14 | **61.62** ±0.34 | **17.38** ±0.18 | **20.69** ±0.18 |
| | ✓ | | | ✓ | 26.21 ±0.17 | 61.67 ±0.36 | 18.23 ±0.17 | 21.83 ±0.19 |
| | ✓ | ✓ | ✓ | | 20.46 ±0.14 | 58.59 ±0.35 | 14.73 ±0.16 | 17.24 ±0.16 |
| | ✓ | ✓ | | ✓ | 22.95 ±0.17 | 58.58 ±0.35 | 16.06 ±0.08 | 19.00 ±0.17 |
| Mask R-CNN+SAM+DSM | ✓ | ✓ | | ✓ | **25.94** ±0.12 | **61.19** ±0.17 | **17.73** ±0.14 | **21.36** ±0.10 |
| | ✓ | ✓ | ✓ | | 20.47 ±0.13 | 58.16 ±0.20 | 14.49 ±0.11 | 17.05 ±0.12 |
| | ✓ | ✓ | | ✓ | 22.83 ±0.09 | 58.15 ±0.20 | 15.77 ±0.09 | 18.71 ±0.08 |

Table 15: Comparison of using different mask and box prompts and scores for the Mask R-CNN+SAM-based models on the SBL test set. We highlight the best combination of prompts and scores in **bold** for Mask R-CNN+SAM and Mask R-CNN+SAM+DSM.

## D.3 Incorporating DSM information

We report results for different Mask R-CNN-based models incorporating DSM information on the SBL test set in Table 16.

To obtain DSM gradients, we used `numpy.gradient` with spacing 1 to get vertical and horizontal gradient maps. Some tiles at the border of AOIs have black pixels, which would lead to very high gradient values between the black areas and the image area. In this case, we paste a mask of of zeros, covering to the black pixels area, onto the DSM gradient maps.

For the model with extra capacity in the Faster R-CNN head of Mask R-CNN, we added an extra Linear layer followed by ReLU activation before the output layers of the bounding box predictor and the classifier of Faster R-CNN.

For the model with an added DSM encoder, the DSM encoder architecture is 3-layer CNN with layer normalizations and GeLU activations. The CNN layers are defined as following:

- First layer: Kernel size $(2, 2)$, with 192 output channels, and same padding.

- Second layer: Kernel size $(2, 2)$, with 768 output channels, and same padding.

- Third layer: Kernel size $(1, 1)$, with 1 output channel.

The output is the same size as the original DSM. This setup was used on the SBL and BCI datasets. Note that the Mask R-CNN+DSM encoder models were trained on a single GPU with 80G GPU memory and 48G CPU memory.

| | Single-class | | Multi-class | |
|---|---|---|---|---|
| Model | mAP | mIoU | mAP | wmAP |
| DSM | 32.37 ±0.18 | 64.08 ±0.17 | **20.87** ±0.13 | 26.82 ±0.15 |
| DSM gradients | **32.43** ±0.41 | 64.55 ±0.35 | 20.68 ±0.17 | 26.91 ±0.29 |
| Extra capacity in Faster R-CNN head | 32.37±0.26 | 64.55 ±0.35 | 20.82 ±0.08 | **26.95** ±0.16 |
| DSM encoder | 32.35 ±0.17 | **64.80** ±0.20 | 20.54 ±0.20 | 26.76 ±0.16 |

Table 16: Results for different Mask R-CNN-based models incorporating DSM information on the SBL test set. Metrics are multiplied by $10^2$ and reported with standard errors. We highlight the best model for each metric in **bold**.

## D.4 Losses

### D.4.1 Hierarchical loss

We define a hierarchical loss, modifying it from [63] since we consider different classes of interest. Similarly to [63], we consider 3 losses, "species", "genus" and "family"-level. When computing the species loss, we exclude instances that have ground truth labels in [Betula, Acer, Magnolopsida,Pinopsida]. In other words, only instances that have a species-level label or that do not have any subcategory at the species-level in the annotations contribute to the loss. For example, we include Picea as a class contributing to the species loss, because there is no class that corresponds to a finer Picea species-level. When computing the genus level loss, exclude instances that have ground truth labels in [Magnolopsida,Pinopsida]. We use the same weights as [63] for species, genus and family losses in the final loss.

## D.5 Results for different losses

We summarize results for Mask R-CNN models trained with different losses in Table 17. Hierarchical loss improves slightly but not significantly on the cross-entropy used in all our experiments.

| | Single-class | | Multi-class | |
|---|---|---|---|---|
| Loss | mAP | mIoU | mAP | wmAP |
| Cross-entropy | 32.44 ±0.12 | **65.08** ±0.44 | 21.38 ±0.17 | 27.27 ±0.18 |
| Weighted loss | 13.23 ±0.17 | 62.05 ±0.17 | 8.10 ±0.18 | 12.32±0.31 |
| Hierarchical loss | **32.76** ±0.16 | 64.70 ±0.25 | **21.42** ±0.15 | **27.51** ±0.10 |
| Focal loss | 30.79 ±0.39 | 64.55 ±0.25 | 14.63 ±0.21 | 23.49 ±0.30 |

Table 17: Results for different Mask R-CNN the SBL test set using different losses. We highlight the best model for each metric in **bold**.

