# OpenReview forum: "Bringing SAM to new heights: leveraging elevation data for tree crown segmentation from drone imagery"
_NeurIPS.cc/2025/Conference — NeurIPS 2025 poster_

### Official Review · Reviewer_QGDS · 2025-06-24

**Clarity:** 4
**Significance:** 3
**Originality:** 3
**Rating:** 5
**Confidence:** 4

**Summary:**

The paper presents a tree crown segmentation approach based on SAM, adapted into a dedicated architecture. The model is evaluated across three datasets with diverse biomes. The authors incorporate photogrammetry-derived DSM as an additional modality and conduct extensive experiments and ablations. The study concludes that SAM out-of-the-box is ineffective for this task, prompt learning dramatically improves performance, and adding DSM information results in marginal but consistent improvements.

**Questions:**

**Questions:**
- To run an RGB-only version of BALSAM, consider running experiments with VHR canopy height prediction models such as:
  - [1] *Depth Any Canopy*, ECCVW 2024
  - [2] *Very high resolution canopy height maps from RGB imagery*, RSE 2024
  - [3] *Open-Canopy*, CVPR 2025
- Can the photogrammetry pipeline to recover the DSM be elaborated further?
- Weighted entropy loss shows no benefit; alternative losses like focal loss could be tested, possibly integrated into the hierarchical loss.

**Remarks/Comments:**
- The authors should clarify the terms of "pretrained" across tables and models to avoid misleading readers, as it refers to a specific part of a specific model, and can seem misleading as all models are pretrained to various degrees.
- The figures would be so much better in vector format.
- Tables are hard to read. Remove excessive digits, inconsistent spacing, and unnecessary parentheses around standard deviations.

**Ethical Concerns:**

["NO or VERY MINOR ethics concerns only"]

**Final Justification:**

The paper is interesting and original, and my last concerns were addressed.

**Quality:**

3

**Strengths And Weaknesses:**

**Strength:**

- Clear and well-written manuscript.
- Extensive and well-designed experiments.
- Thorough ablation
- Evaluation on three datasets across diverse biomes.
- Fair and appropriate baselines
- Thematic and insightful analysis of results.

**Weaknesses:**

- The splitting strategy for the Quebec Plantations dataset may introduce spatial autocorrelation due to training and testing on the same sites; a site-wise split would be more robust.
- The DSM modality only brings marginal improvements (~0.5%) and does not enable the model to consistently outperform simpler alternatives like RSPrompter.
- The model comparisons are somewhat unfair, as not all baselines use the DSM; a canopy VHR height estimation model could have been used for RGB-only scenarios (e.g., [1,2,3]). At least the authors are very clear on which model has access to what

**Overall opinion:**

The reviewer appreciates the solid effort and thorough evaluation, although the core contributions and findings are not particularly ground-breaking. The study is methodologically sound and offers useful empirical insights, especially regarding the limited benefit of DSM and the effectiveness of prompt learning. Overall, the reviewer likes the paper for reporting solid, honest, not exactly ground-breaking but ultimately useful results on a difficult and overlooked task.

---

> ### Author Rebuttal · Authors · 2025-07-30
>
> Dear reviewer,
>
>
> Thank you for your thoughtful review and helpful comments.
> We are pleased to hear that you appreciate the quality of this work and the importance of the problem we are tackling  and that you highlighted that you _“[like] the paper for reporting solid, honest, not exactly ground-breaking but ultimately useful results on a difficult and overlooked task.”_
>
>
> We respond to your comments and questions below.
>
>
> **W1- Splits:** We will make the explanation on splits clearer in the paper. First, we would like to clarify that there are neither image pixels, nor annotations that belong to two different splits. Furthermore, while some sites are represented in both the testing and training sets in the Quebec Plantations dataset, we tried as much as possible to assign sites fully to splits and limit spatial autocorrelation. The only reason why some orthomosaics were divided into both the training and test sets is the distribution of species across sites. Indeed, we wanted to make sure that species of interest were both in the train and test set. This is the only time when we manually drew AOIs. However, as much as possible, the train/val/test regions were kept as spatially separate “blocks” ensuring no overlap between the splits. The table below shows in which splits the sites are assigned. Only the sites afcagauthmelpin and afcagauthier were broken up into the train and test sets. We will add a visualization in the Appendix of the splits for both the training and testing sets.
>
>
> |Site  | **Train** | **Val** | **Test** |
> |---|---|---|---|
> | cbpapinas |  | x | x |
> | cbblackburn1 |  | x |  |
> | cbblackburn2 | x |  |  |
> | cbblackburn3 |  |  | x |
> | cbblackburn4 |  |  | x |
> | cbblackburn5 | x |  |  |
> | cbblackburn6 | x |  |  |
> | cbbernard1 | x |  |  |
> | cbbernard2 |  | x |  |
> | cbbernard3 |  | x | x |
> | cbbernard4 | x |  |  |
> | afcamoisan | x |  |  |
> | afcahoule | x |  |  |
> | afcagauthmelpin | x | x | x |
> | afcagauthier | x | x | x |
>
>
> **W3/Q1-Q2 - re an RGB-only version of BalSAM  and explaining the photogrammetry pipeline to recover the DSM** :
> We understand your suggestion of running an RGB-only version of BalSAM as a suggestion to use canopy height estimation from RGB images instead of the DSM as input to the model. We would like to clarify that the DSM is already a product derived from RGB imagery only, and is always available at no additional cost if drone RGB imagery is available. As a consequence, it is also much higher resolution than the imagery used by the models in the papers linked in your review. For example, the DSM on the Plantations dataset is 0.5cm/pixel resolution, while HRV canopy height maps are typically ~0.3m.
>
> Structure-from-motion (Sfm) photogrammetry, which is used to create RGB orthomosaics from high-resolution RGB drone imagery, generates 3D photogrammetry dense point clouds, from which the DSM is derived.
> In essence, the approach is to:
> 1. Find key points in individual images (SIFT).
> 2. Match keypoints in image pairs to estimate camera locations and parameters (SfM).
> 3. Densify the 3D point cloud (MVS).
> 4. Project that point cloud to 2D to get the DSM.
>
> We refer the reviewer to [1] for further details on the photogrammetry pipeline.
>
> Fig 1. of this paper shows that the DSM is just a 2D projection of the dense point cloud, with interpolation to fill missing values when projecting the 3D points to a 2D grid (raster).
> Therefore, in some sense, BalSAM is already based only on RGB imagery. Please let us know if that answers your question.
>
> As for the comment that “model comparisons are somewhat unfair, as not all baselines use the DSM”. Could you kindly clarify your concern? One of the axes of this study is to determine when and how the DSM is useful, which is why we explicitly compared the same methods with and without DSM input, as shown in our results tables 1, 2 and 3 with the checkmarks in the column “DSM”.
>
>
> **W2. “The DSM modality only brings marginal improvements and does not enable the model to consistently outperform simpler alternatives like RSPrompter.”** A main goal of this study was actually to investigate whether the DSM helps for certain forest contexts and architectures. Our study shows 1/ that using the DSM alongside RGB imagery improves segmentation and classification mean results in the case of tree plantations, which have a specific structure, 2/ in mature forests, the DSM either improves results or keeps them comparable within error bars, 3/ while SAM out-of-the-box does not perform well, SAM-prompt based methods (RSPrompter, BalSAM), which have so far not been used for tree crown segmentation, reach best performance overall on all the datasets.
> This work is a first step towards guiding practitioners in their choices of models for forest monitoring, where different use cases may call for different methods depending on the nature of the data.
> BalSAM only has an additional 10M parameters compared to RSPrompter, which is a fraction of the total model size (the SAM image encoder alone has 632M parameters).
>
>
> **Q3 – re the focal loss:**
> Upon your suggestion, we ran an **experiment with Focal loss with Mask R-CNN on the SBL dataset** (to be consistent with the setup of results reported in table 14 Appendix D.4.) The gamma parameter of the loss was set to 2. Results are reported below:
> | mAP single-class | mIoU single-class | mAP | wmAP|
> |---|---|---|---|
> |30.6|63.9|14.8|23.7|
>
>
> The model performs better than with the Weighted Loss, but not as well as when using Cross-entropy loss. We are currently running more experiments with other values of the gamma parameter.
>
>
> **Remarks: “[...] clarify the terms of "pretrained" across tables and models [...]”:**
> For R-CNN based models, we explain in the text and tables captions that “pre-trained“ models refer to backbones initialized with ImageNet weights as opposed to trained from scratch. For SAM-based models (SAM, RSPrompter, BalSAM), we indeed use the weights from SAM but the image encoder and mask decoder are kept frozen in all configurations. We’ll make sure to highlight this better in the paper.
>
>
> **Formatting:** Thank you for the very helpful feedback, we took this into account and will change figures to vector format and format the tables in a clearer way.
>
> Many thanks for taking the time to review our paper and please let us know if you have further questions.
>
>
> [1] _Iglhaut J, Cabo C, Puliti S, Piermattei L, O’Connor J, Rosette J (2019) Structure from Motion Photogrammetry in Forestry: a Review. Current Forestry Reports 5: 155–168._

---

> > ### Comment · Reviewer_QGDS · 2025-08-01
> > **Regarding the rebuttal**
> >
> > The reviewer thanks the authors for their response. All previously raised points have been generally addressed, and the reviewer is happy to increase their rating.
> >
> > Regarding the DSM, while it is true that it is derived from the same RGB imagery, it requires multiple images to run the SfM pipeline. This means that using the DSM implies access, albeit indirectly ,to additional information. It would have been interesting to see whether a depth or canopy height prediction model could recover part of the performance gain without access to this extra information.
> >
> > This is not so much a weakness as a suggestion, or at least a point the authors should make especially clear in the text.

---

> > > ### Author Response · Authors · 2025-08-04
> > >
> > > We are pleased to have addressed your concerns and appreciate your valuable feedback. We will clarify this point in the text and mention the potential use of a depth or canopy height model as a promising future direction to explore. Thank you again for your suggestions, which will help us to improve our work.

---

### Official Review · Reviewer_XSSG · 2025-06-24

**Clarity:** 4
**Significance:** 3
**Originality:** 3
**Rating:** 5
**Confidence:** 5

**Summary:**

This paper highlights the challenge of identifying and segmenting individual trees from high resolution imagery across different geographic contexts. To address this challenge, this paper presents an approach, BalSAM, to incorporate elevation data in the form of digital surface models into SAM-based workflows to segment individual trees in UAV imagery. This approach is compared to a range of other segmentation approaches including applying SAM to just UAV images and CNN-based tree segmentation models. The authors demonstrate that their approach, BalSAM, outperforms a regular SAM-imagery based segmentation approaches and custom-trained CNN models.

**Questions:**

Fig. 7 presents examples of prompts generate by local maxima in DSMs, many of these point prompts do not fall on trees. I was wondering if the author’s filtered out points that are obviously not on / in the centre of tree canopies using sensible height thresholds?

The ablation studies section could be summarised and point to relevant sections in the appendix and a suggestion would be to explaining / discussing challenging cases (e.g. BCI data) and outlining avenues for future research that flow from this work.

What cases are causing error in the model for the different datasets? Is it small trees? Is it resolving the shapes of complex crowns / overlapping trees? It would be good to include some information on this in the results / discussion sections.

What are the author’s thoughts on how application-ready the BalSAM method is for the use cases outlined in the introduction that need individual tree segmentations with species labels? It would be good to include comments on this in the conclusion section.

**Ethical Concerns:**

["NO or VERY MINOR ethics concerns only"]

**Final Justification:**

I stand by my initial recommendation that this paper be accepted.

**Limitations:**

Yes.

**Quality:**

3

**Strengths And Weaknesses:**

The motivation for the paper was clearly articulated -  the challenge of segmenting and classifying individual trees from UAV imagery and the need for this information. The authors have done an excellent job clearly presenting this work; it is well written and the easy to follow the methods and results. There is a thorough comparison of the main approach developed here for integrating DSM’s into RGB image-based segmentation of individual trees, BalSAM, with a range of different models. It was good that the authors evaluated the different individual tree segmentation approaches across a range of geographic contexts.

The discussion and ablation sections were quite general. There was an opportunity to provide explanation of the results and findings presented in this paper, to discuss considerations for broader application of the BalSAM approach, and to suggest avenues for further research. The authors note the challenges of segmenting individual trees in the dense tropical forest case. It would have been good to hear the author’s thoughts on how to move forward in these cases and to possibly consider opportunities offered by different classification / modelling approaches or ways of summarising information in these scenes for downstream tasks (e.g. estimating carbon or species abundance within larger quadrats / grid cells). It would have been good for the authors to offer suggestions on how to generalise the BalSAM approach to locations where labels are not available for fine tuning.

---

> ### Author Rebuttal · Authors · 2025-07-29
>
> Dear reviewer,
>
>
> Thank you very much for your insightful comments and valuing the importance of this application in forest monitoring and the potential for impact of our work.
>
> We respond to your questions below.
>
> **Q1–Local maxima prompts:**  While there are indeed many ways to manually refine rules for filtering local maxima, we did not do any filtering. We originally experimented with setting thresholds on the DSM or tuning further the size of the neighbourhood to define local maxima. However, there is tall herbaceous vegetation in some sites, and well-tuned parameters on a site would not necessarily transfer well to other sites within the same dataset. Also, the DSM does not provide height with respect to the ground but rather relative height between objects in the image, and does not account for differences in terrain elevation. While it could be possible to normalize the DSM with respect to the lowest point in a site, if we have imagery from a site on a very angled slope, filtering local maxima based on a threshold would not necessarily bring much improvement. Moreover, having too many DSM prompts on tree crowns even if they are not in the center is not the main limitation of this method because each point is fed independently to SAM. If segmented objects overlap, we use NMS and only the object with highest confidence score is kept.
>
>
> **Q2–Discussing challenging cases and directions for future research:**
> Thank you for your suggestions. Indeed, the BCI dataset is especially challenging because of the structure of the forest, including overlapping crowns; the lower resolution and quality of the imagery; and the long-tailed class distribution as highlighted in  5.1 and figure 4). We will provide further details and illustrative visualizations of predictions on the different datasets in the paper. Please see our response to Q3 for more on this topic, which is related to analysing errors of the models.
> Thank you also for your suggestions for improving the discussion section. We will provide a more comprehensive discussion on directions for future research, expanding on the ideas presented in the conclusion (L359-372). In particular, regarding exploring other ways of using the DSM, for the R-CNN based models, one could  consider models using DSM prediction as an auxiliary task rather than as an input to the model.
> A different avenue is making use directly of the 3D point clouds that are used to obtain both the DSM and the RGB orthomosaics.
>
> We also believe that adding contextual metadata into our models in the form of spatial information, spectral information, and the type of forest could improve the performance of tree crown instance segmentation models.  In fact, while we have so far trained models separately on each dataset,  as more high-resolution drone imagery data becomes available, it could be very interesting from an application perspective to learn representations on combined datasets at different resolutions, and location metadata could prove helpful, as it has been shown to in species distribution modelling contexts. Spectral information, available through e.g. satellite open-source programs, could also be added as input into the models even if it is not available at a resolution as high as that of the drone imagery, as additional signal.
>
>
> **Q3- “...what is causing error in the model for the different datasets”**
> We already provided a number of elements for analysis on the Quebec plantations dataset but will make sure that we include more details and  additional visualizations on other datasets as well. Regarding the Quebec plantations dataset, as illustrated in row 1 of Fig. 3, a case that is consistently missed across models is presented when trees crowns are very close to each other or don’t stand out much from the background. We have also observed errors in sites with tall herbaceous vegetation. As highlighted in the discussion, some classification errors align with human errors (e.g. distinguishing white and black spruces)
> Across datasets, overlapping crowns are indeed challenging. As an ablation, we trained a Mask R-CNN with DSM inputs only (no RGB image) to assess its informativeness on the Plantations dataset vs the SBL dataset.
> | **Dataset**   | **mAP single class** | **mIoU** | **mAP**
> | ---- | ----|----|----|
> | Plantations  | 0.464 |0.734 | 0.178
> | SBL | 0.092 |0.289 |0.023
>
> When looking at the drop in performance compared to the Mask R-CNN models using RGB images as inputs, we observe a much larger drop on the SBL dataset compared to the Plantations dataset. In fact, single-class mIoU remains high, which is another way to see that tree crowns are distinguishable from the background and from each other using the DSM only, and points to the usefulness of the DSM in plantation contexts.
>
> **Q4- application-readiness** – Our study shows that using the DSM along the RGB imagery consistently improves segmentation and classification results for the plantation use case, therefore, we would recommend that practitioners looking to quantify the carbon stored in boreal plantations include the DSM information in their models. We leave it to practitioners to decide, based on their application and available data which models to use. For example, if one only has access to bounding boxes as annotations, we showed that Faster R-CNN+SAM is a reasonable baseline to consider and that including the DSM helped (on both the SBL and Plantations datasets).
> We will improve the discussion section with your feedback and outline more clearly future work directions with elements from our responses to Q2 and Q4 as well.
>
> Thank you again for your time and reviewing our paper, and please let us know if you have any other questions.

---

> > ### Comment · Reviewer_XSSG · 2025-08-03
> >
> > Thank you for your thorough responses. The author's point regarding the DSM not accounting for terrain elevation is important and could be stated in the manuscript to remind readers of this fact. I assume the reason that normalised surface models / ground surface models cannot be generated is due to no gaps in the canopy to see the ground surface in some scenes?

---

> > > ### Author Response · Authors · 2025-08-04
> > >
> > > Thank you for your message. We will indeed state this explicitly in the revised manuscript. The reason that ground surface models cannot be generated is indeed due to the lack of dense enough gaps in the canopy to obtain a map of the ground surface, as well as the potential presence of vegetation of intermediate height between the canopy and ground (depending on the type of forest). For this reason, ground surface models are generally created using other sensors such as LiDAR.

---

> > ### Comment · Reviewer_XSSG · 2025-08-05
> >
> > I was interested in the author's comments regarding including contextual metadata, spectral or spatial information in the models or satellite data to improve tree crown segmentation in dense forests. Are the authors able to provide more specifics on examples of this data, how it could be incorporated into the model, and rationale for using this data to discriminate individual tree crowns in dense canopy. As I understand it, tree crowns cannot be discriminated visually in high resolution RGB or DSM images, how will presumably coarser resolution satellite data assist here? I am curious as to at what point segmenting individual tree crows is not possible and alternative strategies would be more prudent (e.g. going directly to mapping biomass or another biophysical variable as a continuous surface)?

---

> > > ### Author Response · Authors · 2025-08-07
> > >
> > > Thank you for your questions. Indeed, we believe that integrating contextual metadata including forest type, spectral or spatial information could improve tree crown instance segmentation models. As you pointed out, these data usually have coarser resolution than high resolution drone imagery. We do not necessarily expect to leverage these modalities to improve individual tree crown delineation, but rather the classification in mature and dense forests.
> > >
> > > For instance, multispectral contextual metadata from Sentinel 2 (e.g. 1x1 to 5x5 pixel grids), could enhance classification for trees with diverse spectral signatures, such as those in the BCI dataset (10-50m crown diameter).
> > > Regarding spatial metadata, location embeddings have proved useful in a variety of biodiversity and earth monitoring tasks such as species distribution modelling.
> > > Along with forest type descriptions and species range maps, we believe that as more drone imagery data for tree monitoring becomes available, such metadata could be beneficial when training on larger or combined datasets across various biomes and regions.
> > >
> > > All these additional information are georeferenced, they can be aligned with RGB imagery at the pixel level or processed via an auxiliary image-level branch, depending on its spatial resolution. One could imagine a framework in which the  tree crown segmentation task would still be performed with the high resolution imagery while potentially improving the classification with fused additional information.
> > >
> > > We thank the Reviewer for this insightful discussion and remain available to answer any questions.

---

### Official Review · Reviewer_XmBk · 2025-07-03

**Clarity:** 3
**Significance:** 3
**Originality:** 3
**Rating:** 5
**Confidence:** 3

**Summary:**

Accurate individual tree crown detection is a essential component of precision forestry. This work proposes BalSAM, which integrates Segment Anything Model with the DSM elevation information for instance segmentation and species classification of individual tree crowns in UAV images. The paper systematically evaluates the method on three typical forest datasets. The results show that: 1) The out-of-the-box SAM has limited effect; 2) Fine-tuning SAM through prompt learning and introducing DSM can achieve mAP improvements; 3) In dense forests with many species, DSM still brings some segmentation benefits but has little impact on classification.

**Questions:**

Q1: Explanation on the low accuracy of multi-class classification.
Q2: Discussion on latest Transformer-based instance segmentation or fine-tuned SAM.

**Ethical Concerns:**

["NO or VERY MINOR ethics concerns only"]

**Final Justification:**

After reading the authors' response and other reviewers' comments, my previous concerns have been well addressed and I do not have further concerns on this paper. I will raise my score to "Accept"

**Limitations:**

yes

**Quality:**

3

**Strengths And Weaknesses:**

S1. This study is the first to apply SAM to the tree crown instance segmentation task, and integrated DSM height data to improve the segmentation. The proposed BalSAM draws on the segmentation ability of SAM, while achieving cross-modal data fusion by adding height information, and expands the application of SAM in remote sensing.
S2. It covers three representative forest datasets with significant differences, including artificially planted forests, temperate forests with diverse species, and tropical rainforests with high canopy density, and compares multiple baselines such as Mask R-CNN, Faster R-CNN + SAM, etc. The conclusion is representative.
S3. Integrating DSM information has improved the segmentation and classification accuracy of the model, especially in planted forests where tree distribution is regular and canopy separation is distinct. The tree crown contours segmented by BalSAM are more in line with the actual shapes and the model effectively reduces the occurrence of species misclassification.
S4. Digital Surface Model information is derived from RGB drone imagery. It does not require additional sensors and has direct application potential for resource-constrained forestry monitor.
W1. On datasets with multiple species such as tropical rainforests, the multi-class mAP remains at a low level (≤ 9%), and its contribution to fine-grained classification is limited.
W2. The article lacks a comparison with the latest Transformer-based instance segmentation or fine-tuned SAM, and the superiority argument is somewhat insufficient.
W3. The standard error of the result is relatively large, further robustification might be necessary in actual deployment.

---

> ### Author Rebuttal · Authors · 2025-07-29
>
> Dear reviewer,
>
> Thank you for your helpful review and for valuing our contribution to the field of forest monitoring with this first study on applying SAM and integrating DSM information for tree crown instance segmentation.
>
> We respond to your questions below.
>
> **W1/Q1** Indeed, the classification accuracy remains low overall on the BCI dataset. As we briefly mention in the discussion, we think this is due to the higher diversity of trees in this tropical context and to the lower resolution and quality of the images from this dataset compared to the other datasets. Additionally, annotations of the BCI dataset are noisy (as can be seen in Figures 1 and 3) and some tree crowns are not annotated. This reflects the complexity of the task of delineating their crowns and classifying them, even for experts in biology. The examples in Figure 3 (rows 3 and 4) show that our models sometimes segment trees that were not labeled. This may also lead to an underestimation of the performance of the methods.
> A potential avenue for tackling this challenge is to continue the work on using different losses that we present in D.3, as well as to explore more losses suited for long-tailed distributions.
>
> **W2/Q2** Upon your suggestion, **we conducted experiments with Mask2former using the Swin-base and Swin-L backbones on the Quebec Plantations dataset** using RGB or RGB+DSM as input. We report results in the table below.
> The models were initialized with COCO-pre-trained weights, trained for a maximum of 100 epochs (as were the other models presented in the paper), with batch size 8 and took ~1.5 days to train on a single GPU. Given the timeline of the rebuttal,  we performed only minimal HP tuning on the learning rate and learning rate scheduler. We find that the DSM improves performance for the Mask2Former Swin-base model. It is possible that results could be further improved, and that models using the larger Swin-L backbone require longer training. Therefore, we are currently conducting further HP tuning with Mask2Former and will add results on multiple seeds in the main paper. That being said, several previous works have highlighted limitations of Mask2Former in the context of forest monitoring from remote sensing imagery.
>
> For example, in [1],  the authors highlight:  _“There are several transformer-based successors to Mask-RCNN, from the DETR family of models to Mask2Former. While we experimented with these architectures, [...], we were unable to reliably outperform Mask-RCNN and found poorer convergence behaviour”_.
> [2] also find that  Mask2Former models’ _“performance is not comparable to the CNN-based architectures”_ in the context of semantic segmentation on the SBL dataset.
>
> | **Model**   | **mAP single-class** | **mIoU single-class** | **mAP** | **wmAP**
> | ---| ---|---|---|---|
> | Mask2Former Swin-base RGB| 55.2|70.0|34.3|37.5|
> | Mask2Former Swin-base RGB+DSM| 58.1|71.9|37.1|39.6|
> | Mask2Former Swin-L RGB| 62.7|74.7|45.1|46.0|
> |Mask2Former Swin-L RGB+DSM|62.1|73.8|42.0|44.8|
>
> Regarding the suggestion to fine-tune SAM, this would unfortunately not solve the problem of instance segmentation - in which we also care about classifying the segmented instances, since SAM predicts masks without classes. Moreover, SAM’s image encoder has 632M parameters and fine-tuning SAM would require considerable compute resources, rendering its training process inaccessible to most forest-monitoring practitioners. In order to train RSPrompter and BalSAM on a single GPU, we used a batch size of 2, while fine-tuning SAM with limited computational resources would simply not be feasible.
>
> **W3**: “The standard error of the result is relatively large, further robustification might be necessary in actual deployment.” We acknowledge that this is a limitation of RSPrompter and BalSAM, as we wrote in L360-361, and agree that further robustification is a priority for future work. This could be done by stabilising the training process through regularization and augmentations, especially on hard classes. We will also further analyse the errors and success cases in the best and worst runs from different seeds of the same model to better understand where issues arise and inform the choice of regularization and augmentations.
>
> We hope that this answers your questions, and please let us know if you need any further clarifications.
>
>
> [1] _Veitch-Michaelis, J., et al. "OAM-TCD: A globally diverse dataset of high-resolution tree cover maps." Advances in neural information processing systems 37 (2024)_
>
> [2] _Ramesh, V., et al. "Tree semantic segmentation from aerial image time series." (2024)_

---

> > ### Comment · Reviewer_XmBk · 2025-08-06
> >
> > After reading the authors' response and other reviewers' comments, my previous concerns have been well addressed and I do not have further concerns on this paper. I will raise my score to "Accept".

---

> > > ### Author Response · Authors · 2025-08-06
> > >
> > > Dear reviewer,
> > >
> > > We are glad to hear that we addressed your concerns. Thank you for your valuable feedback that will help us improve our work.

---

### Official Review · Reviewer_6LSe · 2025-07-03

**Clarity:** 3
**Significance:** 1
**Originality:** 1
**Rating:** 3
**Confidence:** 4

**Summary:**

This paper investigates the application of the Segment Anything Model (SAM) for tree crown instance segmentation from high-resolution drone imagery. The authors introduce BalSAM, which utilizes the RSPrompter framework to incorporate Digital Surface Model (DSM) data for improved segmentation. The study benchmarks several methods—including out-of-the-box SAM, prompted SAM, Mask R-CNN, and RSPrompter—across three distinct forest environments: boreal plantations, temperate forests, and tropical forests. The findings suggest that while off-the-shelf SAM performs poorly, methods that learn to prompt SAM (RSPrompter, BalSAM) marginally outperform a conventional Mask R-CNN baseline. The integration of DSM data is shown to be marginally beneficial, particularly in the structured context of plantations.

**Questions:**

1. The core innovation of BalSAM is the addition of a DSM encoder to RSPrompter. Given that BalSAM does not consistently outperform RSPrompter across all datasets and metrics, how do the authors assess the overall usefulness of DSMs? Why could adding DSM information degrade performance, as seen on the BCI dataset?
2. Could the authors provide a more direct analysis of the trade-offs between accuracy, inference speed, and training cost? Without this, it is difficult to judge whether the modest mAP improvements justify the substantial increase in computational resources when using SAM and RSPrompter.
3. Could the authors at least compare with pretrained Mask2Former, which seems to be a fairer baseline than MaskRCNN, given that the model being evaluated is transformer-based SAM?

**Ethical Concerns:**

["NO or VERY MINOR ethics concerns only"]

**Final Justification:**

After the discussion, my questions regarding the trade-offs between accuracy, inference speed, and training cost have been addressed.

However, I'm not sufficiently convinced that using DSM to prompt SAM is an effective way to fuse DSM information into the model, as the evaluation results are so close to, if not even worse than, the baseline. The authors acknowledge that the evaluation numbers may not fully reflect the model's performance, as certain mistakes in crown segmentation can have a large influence on downstream tasks like AGB estimation. The current experiments do not demonstrate such advantages of BalSAM through additional case studies of downstream applications.

**Limitations:**

The authors largely ignored the limitations in the wide applicability of the proposed method due to the limited availability of high-resolution DSM.

**Quality:**

2

**Strengths And Weaknesses:**

### Strengths
Tree crown delineation problems are well-motivated and significant problems in forest monitoring.

### Weaknesses
1. I have deep concerns over the significance of the results: The main hypothesis of the work is that DSM can help tree crown segmentation. However, the results presented in Section 4.2 suggest that adding DSM has limited improvement, if not the opposite. For example, in Table 1 (test results on the Quebec Plantations dataset), BalSAM achieved an mAP of 65.03 while RSPrompter achieved 66.37. Similarly, in Table 2 (multi-class), RSPrompter is better than the proposed BalSAM.
2. Mask R-CNN and Faster R-CNN with a ResNet backbone are unfair comparisons to SAM, which is based on ViTs with a much larger number of parameters. I encourage the authors to perform additional evaluations with models like Mask2Former [1].
3. I also have deep concerns over the originality of this work. As both SAM and RS-Prompter are previous research, this work's contribution is about adding DSM data into the RS-Prompter pipeline, which is not always helpful. To improve the originality of this work, I encourage the authors to include more discussions around the useful signals to perform tree crown segmentation and provide more insights into the failure points of existing methods, including the proposed BalSAM.

[1] Cheng, B., Misra, I., Schwing, A.G., Kirillov, A. and Girdhar, R., 2022. Masked-attention mask transformer for universal image segmentation. In Proceedings of the IEEE/CVF conference on computer vision and pattern recognition (pp. 1290-1299).

---

> ### Author Rebuttal · Authors · 2025-07-29
>
> Dear reviewer,
>
> We thank you for your time and detailed review, and recognizing the importance of this forest monitoring application.
>
> The goal of this work is two-fold: investigating the task of tree crown instance segmentation with respect to **(i) whether the DSM helps for certain forest types and architectures**, and **(ii) when and how SAM is useful**. Both questions have not been explored extensively before and our work is the first to compare models across those two axes of analysis, even though SAM is the most readily accessible model to many practitioners and the DSM is a product that is always readily available at no additional cost from the RGB imagery.
>
> We show **1/ that using the DSM alongside RGB imagery improves segmentation and classification mean results in the case of tree plantations, which have a specific structure, 2/ in mature forests, the DSM either improves results or keeps them comparable within error bars, 3/ while SAM out-of-the-box does not perform well, SAM-prompt based methods (RSPrompter, BalSAM), which have so far not been used for tree crown segmentation, reach best performance overall on all the datasets.**
>
> Our work is a first step towards guiding practitioners in their choices of models for forest monitoring, where different use cases may call for different methods depending on the nature of the data.
>
> We would also like to highlight that **what might seem like small average differences in evaluation scores can translate to large differences in downstream applications**. Consider the equations used in [3] to determine the amount of carbon stored in trees of different species of the form y_i = beta1_i* (diameter^beta2_i)*(height^beta3_i) where y_i is the biomass of part i of a tree (i = foliage, bark, …). The total biomass of the tree is the sum of the y_i. If one considers y_foliage, the coefficients (beta1, beta2, beta3) are (2.5, 2.45, -2.3) for sugar maple trees and (0.2, 2.38, -1.11) for white spruce - thus, making classification mistakes in an instance segmentation task can lead to huge differences in carbon estimates, especially for small trees (which is what we have in the Plantations dataset).
>
> **We respond to your questions and concerns below.**
>
> **Weakness1/Q1/Limitation**:  While it is true that BalSAM does not always offer a significant advantage over RSPrompter,  our results show that BalSAM either outperforms or performs similarly (within error bars) to RSPrompter across use cases.  As outlined in the discussion (5.1), a reason for the lesser informativeness of the DSM on the BCI dataset could be the complex structure of canopies in tropical forest in which there is high overlap between tree crowns and species diversity,  than in the Quebec Plantations dataset.
>
> Regarding **assessing the overall usefulness of the DSM**, our study shows that the DSM proves useful in most of the experiments when added to RGB information. When the DSM is not helpful (e.g. Mask R-CNN baseline on the SBL dataset), this seems to be due to the structure of the forest which practitioners can assess (see examples of DSMs in figure 1).
> We recommend practitioners to use the DSM along RGB images for plantation use cases where we showed that it consistently improves performance for all methods in terms of the multi-class mAP and wmAP scores.
> We then leave it to users to decide which method is more suited for other use cases, based on which data they have available. For example, if one only has access to bounding box annotations to train their model, the Faster R-CNN+ SAM method could be used and we showed that adding the DSM in this context could bring significant improvements compared to using RGB only.
>
> Relatedly, we would like to respond to your concern in the “limitations section” about “limited availability of high-resolution DSM”, and clarify that the **DSM is in fact always readily available at no additional cost**  when one acquires drone imagery. Structure-from-motion (Sfm) photogrammetry, which is used to create RGB orthomosaics from high-resolution RGB drone imagery, generates 3D photogrammetry dense point clouds, from which the DSM is derived. The 3D point cloud is actually needed for orthorectification to obtain RGB orthomosaic. In other words, the DSM is a by-product of the photogrammetry process and is thus available at no additional cost from RGB drone data processed with Sfm. This is a central point and a key motivation for our study, which we will further emphasize in the paper.
>
> **Q2. “... analysis of trade-offs between accuracy, inference speed, and training cost.”**
>  The table below shows average inference speed per sample for different models using the same V100-SXM2-32GB GPU. While we did not control for the type of GPU used across experiments and therefore training cost figures would be misleading, we  report the number of trainable parameters in each model (this includes the final layer for the Plantations dataset, which has 9 classes). For models that can also take the DSM as a 4th channel along the RGB images as input, the addition of the DSM does not lead to a significant increase in number of parameters.
> | **Model**   | **inference speed** (s/image)| **Number of trainable parameters**
> | --- | ---|---|
> | SAM automatic | 15.3 | 0 |
> | SAM+ height prompts | 1.97 | 0 |
> | Mask R-CNN  | 0.34|43.7M  |
> | RSPrompter | 1.00  |   117M |
> |  BalSAM   | 1.04 |  126M |
>
> We would like to note that during inference time in real world scenarios, we do not have significant constraints (apart from using a single GPU) on the size of the models and their inference speed since the operation is mode offline. Typically, organisations looking into assessing the carbon stored in their plantations do not need embedded system models.
>
> **Weakness2/Q3**. We thank the reviewer for raising the valid concern of comparing models with different capacities and suggesting to add experiments using more recent ViT-based models. **We ran Mask2Former with Swin-base and Swin-L backbones on the Quebec Plantations dataset** using RGB or RGB+DSM as input; we report results in the table below. The models have 106.9M and 215.5M parameters, respectively. They were initialized with COCO-pre-trained weights, trained for a maximum of 100 epochs (as were the other models presented in the paper) and took ~1.5 days to train on a single GPU. Given the timeline of the rebuttal, we performed only minimal HP tuning on the learning rate and learning rate scheduler. We find that the DSM improves performance for Mask2Former Swin-base. It is possible that results could be further improved and that Swin-L models require longer training. We are currently conducting further HP tuning with Mask2Former and will add results on multiple seeds in the main paper. That being said, several previous works have highlighted limitations of Mask2Former in the context of forest monitoring from remote sensing imagery.
> For example, in [1],  the authors highlight: “There are several transformer-based successors to Mask-RCNN, from the DETRfamily of models to Mask2Former. While we experimented with these architectures, [...], we were unable to reliably outperform Mask-RCNN and found poorer convergence behaviour”. [2] also find that  Mask2Former models’ “performance is not comparable to the CNN-based architectures” in the context of semantic segmentation on the SBL dataset.
>
> | **Model**   | **mAP single-class** | **mIoU single-class** | **mAP** | **wmAP**
> | ---| ---|---|---|---|
> | Mask2Former Swin-base RGB| 55.2|70.0|34.3|37.5|
> | Mask2Former Swin-base RGB+DSM| 58.1|71.9|37.1|39.6|
> | Mask2Former Swin-L RGB| 62.7|74.7|45.1|46.0|
> |Mask2Former Swin-L RGB+DSM|62.1|73.8|42.0|44.8|
>
> **Weakness3: “...encourage the authors to include more discussions around the useful signals to perform tree crown segmentation and provide more insights into the failure points of existing methods”**
> Thank you for your helpful feedback. We will expand on the discussion and future work to cover these topics more comprehensively, as follows:
>
> The usefulness of the DSM is (unsurprisingly) linked with the structure of the canopy. Models also tend to perform less well in contexts in which we have overlapping crowns.  As an ablation, we ran a **Mask R-CNN with only the DSM** as input (no RGB imagery) on the Plantations and SBL datasets and report results below. We see a much larger drop in mIoU on the SBL dataset when comparing to Mask R-CNN models using image inputs, while on the Plantations dataset, mIoU remains high, which highlights the informativeness of the DSM in the plantations case.
>
> | **Dataset**   | **mAP single class** | **mIoU** | **mAP**
> | ---- | ----|----|----|
> | Plantations  | 46.4 |73.4 | 17.8
> | SBL | 9.2 |28.9 |2.3
>
> We also believe that a promising avenue for future work is adding contextual metadata into our models in the form of spatial information, spectral information, and the type of forest, to improve the performance of tree crown instance segmentation models. While we have so far trained models separately on each dataset, as more high-resolution drone imagery data becomes available, it could be very interesting from an application perspective to learn representations on combined datasets at different resolutions. Location metadata could also prove helpful, as it has been shown to in species distribution modelling contexts. Spectral information, available through e.g. satellite open-source programs, could also be added as additional input signal into the models even if it is not available at a resolution as high as that of the drone imagery.
>
> We hope this addresses your concerns and we are happy to answer any further questions you may have.
>
> [1] _Veitch-Michaelis, J., et al. "OAM-TCD: A globally diverse dataset of high-resolution tree cover maps."(2024)_
>
> [2] _Ramesh, V., et al. "Tree semantic segmentation from aerial image time series." (2024)_
>
> [3] _Lambert, M.-C., et al. "Canadian national tree aboveground biomass equations." (2005)_

---

> > ### Comment · Reviewer_6LSe · 2025-08-06
> >
> > I thank the authors for the detailed answers. My questions regarding the trade-offs between accuracy, inference speed, and training cost have been addressed.
> >
> > I generally agree that elevation information, like a DSM obtained from SfM, could be useful in tasks like tree crown delineation and AGB estimations. However, I'm not sufficiently convinced that using DSM to prompt SAM is an effective way to fuse such information into the model, as the evaluation results are so close to, if not even worse than, the baseline. The authors acknowledge that the evaluation numbers may not fully reflect the model's performance, so I encourage the authors to find practical downstream use cases, such as AGB estimation, that can clearly demonstrate BalSAM's advantages over the baseline.

---

> > > ### Author Response · Authors · 2025-08-06
> > >
> > > Dear reviewer,
> > >
> > > We thank you for your response and are glad to hear that we addressed some of your concerns.
> > >
> > > Thank you for your suggestion. As we highlighted in the paper and our rebuttal, tree crown instance segmentation is in fact one step towards recovering information on the AGB (of the trees). We understand this work could indeed be extended to other applications and datasets. However, the focus of our study here is high-resolution drone imagery of forested areas.
> > >
> > > For certain contexts (plantations), we see modest but clear improvements with BalSAM, while for other forest types, we don't. We believe that this result in itself is meaningful for applications in forest monitoring. As alluded to in the directions for future work, exploring other ways to encode and incorporate the DSM information is a main avenue for improving our models.

---

### Note · Authors · 2025-08-15

We would like to thank all the reviewers for their feedback, and for the insightful discussions that helped improve our submission. Thank you for recognizing the usefulness of our work on the challenging task of tree crown instance segmentation, and for acknowledging the importance of this forest monitoring application, which is particularly relevant for accurate biomass estimation in the contexts of climate change mitigation and biodiversity preservation.

In this paper, we investigate when and how the Segment Anything Model is useful for the task of tree crown instance segmentation in high-resolution drone imagery, and the potential integrating the Digital Surface Model information into models, which is available at no additional cost from the drone RGB  imagery.  Our work is the first to compare models across those two axes of analysis, even though SAM is the most readily accessible model to many practitioners and the DSM is a product that is always readily available at no additional cost from the RGB imagery. We consider three different forest contexts (boreal plantations, temperate forest, tropical forest) and compare R-CNN-based methods, approaches using SAM out-of-the-box, and methods efficiently-tuning it further—RSPrompter and BalSAM, a method we build on RSPrompter, that leverages the DSM information.  We find that SAM out-of-the-box performs poorly on this task, but our experiments demonstrates the potential of tuning SAM through prompt learning and integrating the DSM into models. We also highlight challenges in dense forests with high species diversity.

During the rebuttal phase, following the reviewers' suggestion, we conducted additional experiments with Mask2Former. Our results are in line with previous works that found Mask2Former did not outperform CNN-based methods in tree segmentation using remote sensing images tasks, and our conclusions on the relevance of RSPrompter and BalSAM remain unchanged. We addressed concerns regarding the availability of the DSM, detailing the process of obtaining it from Structure-from-Motion photogrammetry, which is already used to obtain the orthorectified RGB drone images.
We also outlined more precise directions for future work that we will include in the final version of the paper, in particular, regarding further robustification of RSPrompter and BalSAM and adding contextual metadata into our models in the form of spatial, spectral, or forest type information.

---

### Decision · Program_Chairs · 2025-09-17

**Decision:**

Accept (poster)

**Comment:**

This paper presents a tree crown segmentation approach based on SAM using high-resolution imagery across different geographic contexts. Specifically, the paper proposes BalSAM, which utilizes the RSPrompter framework to incorporate Digital Surface Model (elevation information) data for improved segmentation. The model is evaluated across three datasets with diverse biomes and compared to a range of other segmentation approaches, including SAM, prompted SAM, Mask R-CNN, and RSPrompter—across three distinct forest environments: boreal plantations, temperate forests, and tropical forests (datasets with diverse biomes). The final results show that current baselines perform poorly and that the integration of DSM data shows marginal but consistent improvements, particularly in the structured context of plantations.


Strenghts

S1. The paper is well-written and easy to follow.

S2. The introduction motivates tree crown delineation problems, the significant problems in forest monitoring, and the challenge of segmenting and classifying individual trees from UAV imagery.

S3 The integration of SAM with DSM height to improve tree segmentation. Moreover, both use only RGB drone imagery, so they do not require other sensors.

S4. The experiments were extensive, considering three different geographic datasets and multiple baselines.

S5. The experiments show a slight but consistent improvement with respect to the baselines.


Weaknesses

W1. There are deep concerns about the significance of the results. They are slightly better, but not always consistent.

W2. The inclusion of DSM into SAM is not a very novel work.

W3. The evaluation could be unfair, considering that not all the baselines use DSM.

W4. The baselines could be too complex, and other models could be considered, such as Mask2Former, the latest Transformer-based instance segmentation, or fine-tuned SAM.

W5. The standard errors of the proposed model are relatively large, which may indicate that the results are not statistically significant.

W6. The splitting strategy on Quebec datasets could introduce spatial autocorrelation.

W7. The discussion and ablation sections were quite general. It would have been useful to offer suggestions about the generalization of BalSAM.

Despite significant weaknesses in novelty, experiment fairness, and result significance, most reviewers push for acceptance, with only one reviewer offering a weak rejection. Three reviewers considered that the problem is interesting, and the framework is novel.

The discussion included a question regarding the trade-offs between accuracy, inference speed, and training cost, which was addressed. However, one reviewer was still not convinced that the inclusion of the DSM generates an important contribution to the model. In general, the authors agree with some of the weaknesses, such as significant standard error and slight improvement in the error.